# GenNet: A Generative AI-Driven Mobile Network Simulator for Multi-Objective Network Optimization with RL

## Abstract

Simulation-based optimization has emerged as a crucial methodology in the field of mobile network optimization, addressing the need for dynamic and predictive network management. To address the scarcity of open-source mobile network simulators for advanced research, we developed GenNet—a generative AI-driven mobile network simulator. GenNet can create virtual replicas of mobile users, base stations, and wireless environments, utilizing generative AI methods to simulate the behaviors of these entities under various network settings with high accuracy. GenNet features a tailor-made API explicitly designed for reinforcement learning environments, enabling researchers to finely adjust network parameters such as tilts, azimuth, and transmitting power. Extensive experiments have employed GenNet to benchmark multi-objective optimization algorithms, focusing on enhancing network coverage, throughput, and energy efficiency, validating its effectiveness as a robust platform for advancing network optimization techniques. Through this innovative tool, we aim to empower researchers and practitioners to identify and implement the most effective approaches for network optimization, paving the way for future advancements in mobile network management.

## 1 Introduction

The deployment of the fifth-generation mobile network (5G) has expanded the range of services, users, and devices on mobile networks, resulting in more complex network environments and diverse application scenarios (Wu et al., 2017). This complexity introduces significant challenges for network optimization, which has traditionally depended on mathematical modeling and operations research (Fei et al., 2016; Aliu et al., 2012). In the typical network optimization process, researchers first define control variables and objectives and then model these relationships mathematically before applying optimization algorithms. However, several issues hinder this approach in today's 5G networks: traditional models struggle to capture the physical realities of increasingly complex systems; direct testing of optimized configurations in real-world networks is often not feasible, making it difficult to assess their effectiveness accurately. These issues underscore the urgent need for innovative network optimization strategies to effectively address the dynamic and complex requirements of 5G networks.

Simulation-based optimization has emerged as an innovative methodology for addressing the challenges inherent in current mobile network optimization (Chen et al., 2020; Gong et al., 2023; Huang et al., 2023; Li & Li, 2023). This approach involves constructing a virtual network simulator that acts as a digital twin of a physical mobile network, replicating the structure, environment, and state of network elements or systems with high accuracy. Reinforcement learning-based optimizers work iteratively with this simulation model to identify the most effective network configurations. This technique addresses the two principal challenges of traditional optimization methods: modeling and evaluation difficulties. The simulator avoids complex mathematical modeling by generating a virtual counterpart of the mobile network. Developers and operators can use this simulator to conduct What-if Analysis, allowing them to test various network configurations and optimization strategies without impacting the real-world network. Additionally, because the simulator is designed to be a high-fidelity replica of the real-world network, it ensures that the optimization strategies developed are applicable and effective in real-world scenarios. Thus, simulation-based optimization is poised to

Table 1: Advantages and disadvantages of GenNet and contextualize it within the landscape of existing environments.

| Simulators | Simulation Methods | Platform Language | Realistic Mobility | Realistic Traffic Usage | Link QoS | Scheduling Support | Handover Support | Protocol Stack | Simulation Time Cost | Interpretability |
|---|---|---|---|---|---|---|---|---|---|---|
| Matlab | Rule-based | Matlab | × | × | ✓ | ✓ | × | × | High | High |
| NS-3 | Event-triggered | C++ | × | × | × | ✓ | × | ✓ | High | High |
| OMNet++ | Event-triggered | C++ | × | × | × | ✓ | ✓ | ✓ | High | High |
| OPNET | Event-triggered | C++ | × | × | × | ✓ | × | ✓ | High | High |
| SyntheticNET | Rule-based, Event-triggered | Python | ✓ | × | ✓ | ✓ | ✓ | ✓ | High | Medium |
| CityFlow | Data-driven | Python | ✓ | × | × | × | × | × | Medium | Low |
| MATSim | Data-driven | Python | ✓ | × | × | × | × | × | Medium | Low |
| GenNet | Data-driven, Rule-based | Go, Python | ✓ | ✓ | ✓ | ✓ | ✓ | × | Low | Medium |

play a critical role in the future evolution of mobile networks. It promises to transform the interaction with mobile networks and their operational and optimization models, fundamentally changing the network management landscape.

Creating a virtual simulator of a physical mobile network for performance evaluation is a crucial research area in networking. Network simulators like NS-3 (Henderson et al., 2008), OPNET (Chang, 1999), Matlab (Tariq et al., 2018), and OMNet++ (Köpke et al., 2008) utilize discrete-event-driven simulations to mimic the communication behaviors of network elements and assess performance metrics such as throughput, latency, and data rate. However, these simulators are significantly less efficient (Hui et al., 2022); their simulation speed is much slower than real-world networks. This inefficiency hinders their ability to meet the demands of interactions with reinforcement learning-based optimizers. Hence, there is a pressing need to develop a new virtual simulation technique that can quickly, efficiently, and accurately simulate the dynamic behavior and performance of networks.

In this paper, we propose GenNet, a generative AI-driven mobile network simulator designed for multi-objective network optimization with reinforcement learning (RL). This simulator creates a virtual replica of each physical entity within a mobile network, including mobile users, base stations, and wireless environments. To effectively simulate the behaviors of these network components, GenNet leverages real-world data from mobile networks and employs generative AI methods such as Generative Adversarial Networks (GANs) (Cai et al., 2021), Variational Autoencoders (VAEs) (Cemgil et al., 2020), and diffusion models (Croitoru et al., 2023). Unlike traditional, inefficient discrete-event-driven network simulations, generative AI models in GenNet are trained on extensive datasets from real-world mobile networks. This training enables the models to learn the distributional characteristics of network data under various environmental conditions. This approach facilitates an accurate mapping of environmental factors to network components' behavioral and performance data, resulting in an efficient, AI-driven simulation process.

Importantly, GenNet features a user-friendly Application Programming Interface (API) tailored explicitly for RL environments, making it an ideal tool for RL-based optimizations. As an interactive simulator, GenNet enables researchers to adjust network settings such as mechanical tilt, electrical tilt, azimuth, and maximum transmitting power of base stations and generate corresponding network performance data. Notably, compared to single-objective optimization, which focuses on finding the best solution for a single metric, multi-objective optimization seeks to balance different goals. Multi-objective optimization is essential in mobile networks, where trade-offs must be made between competing factors such as throughput, coverage, energy efficiency, and signal interference. Thus, to facilitate the development and testing of multi-objective optimization RL algorithms in mobile network scenarios, we conduct comprehensive experiments to benchmark multi-objective optimizations in mobile networks. GenNet provides a standardized environment for multi-objective optimization, focusing on optimizing network coverage, throughput, and energy consumption. We introduce RL-based network optimization baselines, a collection of reliable and efficient implementations of state-of-the-art algorithms designed to provide a solid foundation for advancing large-scale mobile network optimization. Notably, all these algorithms are inherently compatible with GenNet.

Our contributions are summarized as follows:

• We propose GenNet, the first open-source generative AI-driven mobile network simulator. GenNet creates virtual replicas of network entities such as mobile users, base stations, and wireless environments. It utilizes generative AI methods to accurately simulate the behaviors of these entities under various network settings based on extensive real-world data, thereby surpassing traditional discrete-event-driven simulations. GenNet is open-sourced and freely available at GitHub[1].

• GenNet provides a tailor-made API for reinforcement learning environments, enabling researchers to adjust network settings like tilts, azimuth, and transmitting power, generate network performance data, and explore and evaluate various optimization strategies to identify the most effective approaches.

• We conduct extensive experiments using GenNet to benchmark multi-objective optimization algorithms in mobile networks to optimize network coverage, network throughput, and energy consumption. We introduce RL-based optimization baselines compatible with GenNet and validate its effectiveness as a robust platform for advancing network optimization techniques.

## 2 RELATED WORK

### 2.1 SIMULATORS FOR MOBILE NETWORK

Among the available mobile network simulators, Matlab (Tariq et al., 2018) is a highly advanced link-level simulator that offers a flexible frame structure and various resource scheduling techniques. However, unlike GenNet, Matlab is not a system-level simulator and lacks features for realistic mobility and traffic usage modeling for mobile users. Since Matlab does not model user mobility, it cannot simulate handover mechanisms. Also, Matlab-based simulators face integration challenges with Python-based reinforcement learning optimizers due to limited platform language support.

Other popular discrete-event mobile network simulators, including NS-3 (Henderson et al., 2008), OMNeT++ (Köpke et al., 2008), and OPNET (Chang, 1999), deliver accurate packet-level results through their comprehensive protocol stack implementations. However, due to the absence of user mobility modeling, NS-3 and OPNET are better suited for core network modeling than mobile access networks, where key performance indicators like coverage and capacity are essential. Although OMNeT++ can integrate predefined user trajectories to model handover mechanisms, the complex and resource-intensive protocol stack simulations make NS-3, OMNeT++, and OPNET computationally demanding, limiting their ability to model large-scale networks with hundreds of elements realistically. Furthermore, they lack APIs for reinforcement learning environments, which restricts their utility in advanced RL optimization scenarios. A recent simulator, SyntheticNET (Zaidi et al., 2020), integrates the advantages of rule-based simulators like Matlab and event-triggered simulators like NS-3, OMNeT++, and OPNET. It uses rule-based methods for link-level flexible frame structure simulation and protocol stack implementation for fine-grained packet-level scheduling. SyntheticNET also leverages SUMO to provide mobile users with realistic mobility. However, SyntheticNET has a much higher computational cost due to its integration complexity.

Besides rule-based and event-triggered simulation methods, simulations also use data-driven approaches. For example, CityFlow (Zhang et al., 2019) and MATSim (W Axhausen et al., 2016) use black-box neural network models to simulate realistic user mobility in a city. One significant advantage of data-driven methods is reduced simulation time. Inspired by this, we designed GenNet, which leverages generative AI methods to simulate mobile networks. Unlike traditional black-box structures, GenNet applies a grey-box approach, decomposing the mobile network into different critical elements. GenNet uses a white-box rule-based method for the connections between elements, while each entity's behavior is simulated using black-box data-driven methods. Benefiting from parallelism, GenNet also excels in simulation time. However, we acknowledge that GenNet has not yet implemented a comprehensive protocol stack, which limits the interpretability of simulation results for internal network elements.

In summary, we compare the advantages and disadvantages of our proposed GenNet and contextualize it within the landscape of existing simulation environments, as presented in the Table 1.

---

[1]https://github.com/xxx, due to double blind review, the full url cannot be given.

**GenNet**

| Mobile User Modeling | Base Station Modeling | Wireless Environment |
|---|---|---|

Trajectory Generation    Traffic Generation    Physical Parameters    Energy Consumption    Physical Environment Modeling    Signal Propagation Modeling

Network Configurations      Network Performance Indicators

**Reinforcement Learning Based Network Optimizer**

Figure 1: The building bocks of GenNet.

## 2.2 LEARNING-BASED MOBILE NETWORK OPTIMIZATION

The application of reinforcement learning (RL) to adaptively configure operational parameters in mobile networks has recently garnered significant research interest. Since the pioneering studies referenced in (Mnih et al., 2015) and (Silver et al., 2016), deep reinforcement learning (DRL) has become a crucial area within machine learning and artificial intelligence. Algorithms such as Q-learning (Watkins & Dayan, 1992), actor-critic methods (Konda & Tsitsiklis, 1999), and policy gradients (Peters & Schaal, 2008) have proven effective in mastering complex tasks in high-dimensional spaces solely through reward feedback. These algorithms have been successfully applied to a wide array of mobile network optimization challenges, including energy optimization (El Amine et al., 2022; Mondal et al., 2021), resource allocation (Huang et al., 2023; Naderializadeh et al., 2021), mobility management (Alsuhli et al., 2021; Marí-Altozano et al., 2021), and power control (Meng et al., 2020; Guo et al., 2020). Despite the potential of learning-based methods to develop network optimization policies, benchmarking these policies on large-scale simulators remains challenging. To support robust and efficient research in mobile network optimization, GenNet provides standardized training and evaluation workflows and reliable benchmarking for both single- and multi-objective optimization tasks. Additionally, we offer implementations of representative RL baseline algorithms and document their performance using a standard set of metrics on GenNet for reference.

## 3 GENNET PLATFORM

In this section, we provide an overview of the building blocks of GenNet and its user interface. A primary objective of GenNet is to base its simulations on real-world mobile network scenarios and model the complex interactions among mobile users, base stations, wireless environments. Additionally, GenNet is designed to be both fast and flexible. Users can easily modify or replace each component described in the building blocks to accommodate their specific project needs.

### 3.1 SIMULATOR BUILDING BLOCKS

As shown in Figure 1, GenNet focuses on three primary components of mobile access networks: mobile users, base stations, and wireless environments. We develop virtual versions of each, which are configured with real-world data. This realism is achieved by modeling their fundamental principles and parameters using generative AI models. By integrating these components, we can then simulate network performance and develop the overall platform. The detailed descriptions of the used modeling methods can be found in Appendix A.1.

**Mobile User Modeling**

Mobile users are pivotal in mobile networks, with their characteristics, such as locations and traffic demands, significantly influencing communication quality. GenNet accurately models and analyzes communication by simulating mobile users' trajectories and traffic demands across various locations and times. The generation of mobile user trajectories in GenNet involves two key steps. Initially, we use ActSTD (Yuan et al., 2022) to simulate user movements between different points of interest, explicitly creating origin-destination trips. This method employs generative adversarial imitation learning to detail user trajectories, including visited locations, visit timestamps, and durations, as illustrated in Figure 2. For example, it can generate a scenario where *user A*

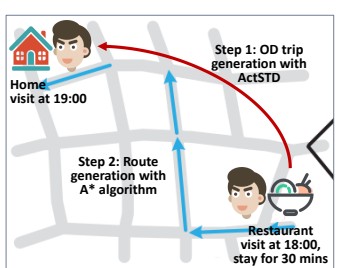

Figure 2: Generating mobile user trajectories with two steps.

visits a restaurant at 18:00, stays for 30 minutes and then proceeds home at 19:00. Subsequently, we employ the $A^*$ algorithm (Hart et al., 1968) to determine the optimal route between these points, marked by blue arrows in Figure 2. This step includes modeling pedestrians and vehicles; pedestrians follow a set speed on sidewalks and crosswalks, while vehicle trajectories are enriched with lane changes and traffic light dynamics using the Krauss model (Krauß et al., 1997). For modeling users' traffic demands, we employ MSH-GAN (Li et al., 2024), a multi-scale hierarchical GAN. This method effectively generates diverse traffic usage patterns using multiple pattern generators and incorporates various switch modes through multiple switch mode generators.

**Base Station Modeling**

GenNet's base stations are set up with real-world physical parameters, such as location, height, mechanical tilt, electrical tilt, azimuth, maximum transmitting power, and the number of antennas. Users can adjust these parameters, but default settings that used by mobile operators are also available. GenNet can simulate mobile networks across the entire metropolitan area of Beijing. The simulation areas encompass a variety of scenarios, including residential, office, entertainment, and transportation areas. Users can adjust the 'microscopic_range' parameter in the 'config.yml' file to select their desired simulation area. In this case, we specifically chose the Guomao area, covering approximately 0.17 $km^2$ and hosting 145 indoor and 39 outdoor base stations, as illustrated in Figure 3. That is because operators typically group approximately 200 base stations into

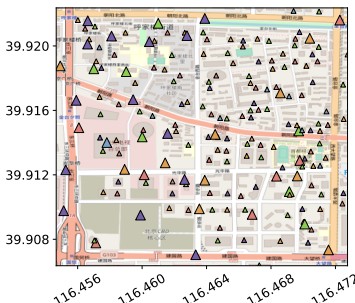

Figure 3: Guomao area along with outdoor (large triangles) and indoor (small triangles) base stations.

a cluster for joint optimization, making the network scale of this area particularly suitable for testing. Also, GenNet models the energy consumption of base stations, which adheres to the methodology proposed by Li et al (Li et al., 2023).

**Wireless Environment Modeling**

The physical environment of a city, particularly the distribution of buildings, significantly affects wireless signal propagation (Andersen et al., 1995). Our study utilizes OpenStreetMap (Haklay & Weber, 2008) to gather detailed information on the buildings in Beijing's Guomao area, including their outlines and heights. Signal propagation models fall into three main categories: stochastic, deterministic, and data-driven. In GenNet, we implement both types. The stochastic model, 3GPP TR 38.901 (Zhu et al., 2021), developed by the 3GPP organization, adapts channel modeling to various settings, including urban, rural, indoor, and outdoor environments. The deterministic model uses Ray Tracing (Yun & Iskander, 2015), which approximates solutions to Maxwell's equations based on the principles of geometrical optics. This method involves launching signal rays from a transmitter and tracing their interactions with the environment, applying theorems of reflection and diffraction to assess changes in signal energy and propagation paths.The data-driven model used PEFNet (Jiang et al., 2024), which integrates computational electromagnetic (CEM) methods with neural networks to enhance realism, efficiency, and generalization.

**Network Performance Modeling**

Network performance encompasses a variety of metrics relevant to mobile devices, base stations, and the overall system. The detailed descriptions can be found in Appendix A.2.

## 3.2 SYSTEM IMPLEMENTATION

GenNet is a mobile network simulation platform that integrates with external optimizers through custom remote procedure calls (RPC). We use MongoDB and PostgreSQL for data storage, as illustrated in Figure 4, which shows the data loading process, simulation, and output. Input data, in-

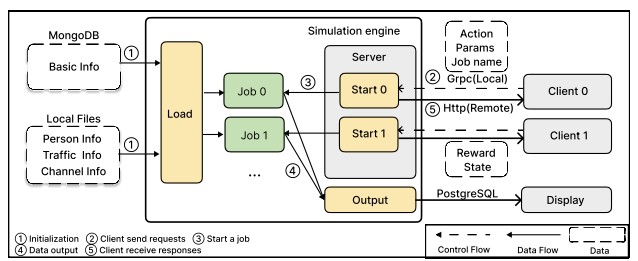

Figure 4: The data flow of GenNet.

cluding base station and mobile user information, is retrieved from MongoDB. Clients initiate

simulations by sending a gRPC or HTTP request containing action information, control parameters, and a job name. Results are stored in PostgreSQL upon completion, and the client receives a response with rewards and status. Our system facilitates network optimization by interacting with RL-based optimizers and supports real-time viewing of optimization results by parallel processing multiple tasks. Simulating millions of mobile users involves complex interactions addressed using parallelization techniques like synchronous updates and mutually exclusive access ((Zhang et al., 2022)). This includes a two-step update process for mobile user interactions with nearby base stations—connection updates and state computation and storage—both spatially parallelizable. Additionally, we enforce mutual exclusivity where only one user can occupy a resource block at any base station, managing this through a mechanism where mobile users connecting to the same base station sequence their interactions. Furthermore, our simulation engine manages concurrent requests efficiently by running multiple coroutines per job name without interference.

### 3.3 ENVIRONMENT INTERFACE

GenNet offers a user-friendly interface that allows researchers to customize network configurations, such as base station transmit power, resource allocation schemes, and antenna settings (azimuth and tilt angles). The platform provides a comprehensive suite of APIs that enable reinforcement learning algorithms to access and modify environmental parameters, optimizing the network setting. A key component is the abstract base class"Env", which outlines essential methods and attributes for a reinforcement learning environment, including state space, action space, environment reset, and action execution. Environments must inherit from this base class and implement its methods, ensuring consistency and functionality across different scenarios. The detailed instructions for the usage of APIs can be found in Appendix A.4.

**"Env.reset (self)"**: resetting the environment to an initial state and returns the initial observation.

**"Env.step (self, action, start, total, interval)"**: accepting the agent's action as input and returns the next state, reward, termination status, and other information. "Action" is the decision action provided by RL. "Start" represents the starting step of the simulator. "Total" denotes the total number of steps the simulator will execute, and "interval" signifies the time interval for each step.

**"Env.render (self)"**: computing the rendering frames specified during the environment's initialization.

**"Env.close (self)"**: closing the environment and frees up resources.

### 3.4 SIMULATION EFFICIENCY

We compared GenNet with popular simulators. All simulations were conducted on a server equipped with 128 GB of RAM, an 8-core Intel Xeon CPU E5-2637 v4 @ 3.5 GHz, and an Nvidia GeForce RTX 2080Ti GPU with 80 GB of memory. Since our simulator integrates multiple key modules into a unified framework, we evaluated individual modules against corresponding state-of-the-art (SOTA) simulators to ensure a fair comparison. Specifically, our simulator consists of two main modules: one for simulating mobile user behaviors and the other for wireless network transmission. For mobile user behavior simulation, we compared GenNet with CityFlow and MATSim. For wireless network transmission simulation, we compared GenNet with Matlab and OMNet++. The experimental results are presented in Table 2. Compared to CityFlow and MATSim, GenNet demonstrates greater computational efficiency due to its parallel processing capabilities. Additionally, when compared to Matlab and OMNet++, GenNet exhibits higher speedup, benefiting from the new data-driven generative AI methods we employed.

Table 2: Simulation Efficiency of GenNet and existing environments.

| Simulators | Simulation Task | Simulation Scale | Computation time |
|---|---|---|---|
| **GenNet** | Mobile user behaviors | 2,464,950 users | **37.70 sec** |
| **CityFlow** | Mobile user behaviors | 2,464,950 users | 3806.7 sec |
| **MATSim** | Mobile user behaviors | 2,464,950 users | 395.48 sec |
| **GenNet** | Wireless transmission | 183 BSs | **1.126 sec** |
| **Matlab (Ray tracing)** | Wireless transmission | 183 BSs | 2.731 hr |
| **OMNet++ (Volume Integral Equation)** | Wireless transmission | 183 BSs | 19.5 min |

# 4 BENCHMARK: MULTI-OBJECTIVE OPTIMIZATION IN MOBILE NETWORKS

We evaluate GenNet as a RL environment by implementing and testing several multi-agent RL optimization baselines designed for multi-objective optimization in mobile networks. We begin by defining the standard multi-objective optimization problem and introducing tailored RL baselines. Next, we conduct an empirical study to compare their performance, aiming to demonstrate GenNet's potential applications through straightforward design choices and configurations. We present benchmark results for 6 baseline algorithms, ensuring reproducibility and reliability by thoroughly testing them on our platform. This emphasizes GenNet's role in facilitating the evaluation of different RL algorithms and provides robust support for multi-objective optimization problems in mobile network scenario. Future research may further enhance the performance of these baseline agents.

## 4.1 PROBLEM DEFINITION

We focus on optimizing antenna angles, beamwidth, and power settings to enhance the performance of a communication network within the Guomao area. We consider three key network performance indicators: network throughput, coverage rate, and energy consumption. Specifically, we aim to maximize the effective coverage rate and user data throughput while minimizing energy consumption.

Network optimization presents a significant challenge, as no single solution can perfectly optimize all objectives simultaneously. For instance, improving the coverage rate (CR) and throughput (TP) may result in increased energy consumption (EC) and operational costs. Balancing these conflicting goals is critical in multi-objective network optimization, which involves a variety of parameters such as antenna angles, transmission power, and beamwidth. The goal is to identify a set of Pareto-optimal solutions, where any improvement in one objective would cause deterioration in at least one other objective. These solutions form what is known as the Pareto front Ngatchou et al. (2005). The multi-objective optimization problem can be mathematically expressed as follows:

$$\max_{\boldsymbol{A}_N} \left\{ \lambda_1 \cdot \mathrm{CR} + \lambda_2 \cdot \frac{\mathrm{TP}}{\mathrm{TP_{max}}} + \lambda_3 \cdot \left( 1 - \frac{\mathrm{EC}}{\mathrm{EC_{max}}} \right) \right\}, \tag{1}$$

$$\text{s.t.}$$

$$P_i \in [0, P_{\max}], \quad \forall i \in N, \tag{2}$$

$$\phi_i \in [0°, 360°), \quad \forall i \in N, \tag{3}$$

$$\theta_i \in [0°, 90°], \quad \forall i \in N, \tag{4}$$

$$\beta_i^{\mathrm{H}} \in [45°, 90°], \quad \forall i \in N, \tag{5}$$

$$\beta_i^{\mathrm{V}} \in [5°, 45°], \quad \forall i \in N, \tag{6}$$

$$\sum_{j=1}^{3} \lambda_j = 1, \quad \lambda_j \geq 0. \tag{7}$$

In the objective function equation 1, the terms $\mathrm{TP_{max}}$ and $\mathrm{EC_{max}}$ are the maximum possible throughput and energy consumption, respectively, used for normalization. The term $1 - \frac{\mathrm{EC}}{\mathrm{EC_{max}}}$ ensures that energy consumption is minimized, as this term will approach 1 when the energy consumption is low. The variables $\boldsymbol{A}_N$ represent the set of actions for base stations (BSs) $N$, including parameters such as transmission power $P_i$, azimuth angle $\phi_i$, tilt angle $\theta_i$, horizontal beamwidth $\beta_i^{\mathrm{H}}$, and vertical beamwidth $\beta_i^{\mathrm{V}}$, subject to the constraints given by equation 2 to equation 6. The weights $\lambda_1$, $\lambda_2$, and $\lambda_3$ in the objective function represent the relative importance of each performance metric: coverage rate, throughput, and energy consumption, respectively. These weights are adjustable and satisfy the constraints equation 7. By changing the values of $\lambda_1$, $\lambda_2$, and $\lambda_3$, we can obtain different optimal solutions, each reflecting a different trade-off among the objectives. The set of all such optimal solutions forms the Pareto front, representing the set of solutions where no single objective can be improved without sacrificing at least one other objective.

Given that the adjustment of antenna angles is limited to a range specific to each optimization step, the decision-making process for the next step is based on the actions of the base stations and the state

of the environment at the previous step. This problem can be effectively modeled as a Multi-Objective Markov Decision Process (MOMDP). A Multi-Objective Markov Decision Process is defined by the tuple $\langle S, A, P, R, \gamma, D \rangle$, where $S$ is the state space. $A$ is the action space. $P(s'|s, a)$ is the state transition probability, representing the probability of transitioning from state $s$ to state $s'$ given action $a$. $R = [r_1, ..., r_m]^T$ is the vector of reward functions, where each $r_i : S \times A \rightarrow \mathbb{R}$ corresponds to a different objective. $\gamma = [\gamma_1, ..., \gamma_m]^T \in [0, 1]^m$ is the vector of discount factors. $D$ is the initial state distribution. $m$ is the number of objectives. In the context of mobile network optimization, the MOMDP is defined as follows. State Space $(S)$ includes all possible configurations of the network, such as the current positions of users, signal strengths, current antenna angles, base station power, and other relevant environmental factors. Action Space $(A)$ consists of all possible adjustments to the antenna angles and base station power. Reward Functions $(R)$ includes multiple reward functions corresponding to the performance metrics we aim to optimize. Discount Factors $(\gamma)$ reflect the importance of future rewards compared to immediate rewards for each objective. Initial State Distribution $(D)$ reflects the starting conditions of the network before any optimization steps have been taken. By modeling the network optimization problem as a MOMDP, we can leverage multi-objective reinforcement learning (MORL) techniques to find Pareto-optimal policies that provide the best trade-offs among the various conflicting objectives.

### 4.2 EXPERIMENTAL SETTINGS

**Experimental Environments**

Experiments are conducted in a densely populated sector of Beijing's Guomao Region. The experimental setup included 183 BSs, comprising 39 outdoor and 144 indoor BSs. It is important to note the disparities in transmission power and path attenuation between indoor and outdoor BSs. The bandwidth allocated for each BS was set at $1.8 \times 10^5$ Hz, and the noise density was maintained at -174 dBm/Hz. The experimental area spanned 1770 meters by 1560 meters, resulting in a grid resolution of 10 meters and a total of 27,612 grids. The threshold for the Reference Signal Received Power (RSRP) was established at -100 dBm. The maximum transmission power of outdoor BSs is 46dBm, and the maximum transmission power of indoor BSs is 24dBm. The other detailed information of experimental settings can be found in Appendix A.3.

**Multi-agent RL Baselines**

We conducted a comparative analysis of the vanilla MAPPO (Yu et al., 2022) and MADDPG (Kaur et al., 2023), two of the most established and widely used multi-agent reinforcement learning algorithms. We also integrated MAPPO and MADDPG with multi-objective optimization methods—Envelope (Yang et al., 2019), a representative single-policy multi-objective optimization approach, and Prediction Guide (PG) (Xu et al., 2020), a typical multi-policy multi-objective optimization technique as benchmark baselines. The detailed introduction of baseline methods can be found in Appendix A.5.

**Evaluation Metrics**

In single-objective RL settings (i.e., MAPPO and MADDPG), policies are evaluated based on their corresponding reward. For multi-objective RL (i.e., Envelope and PG), we utilize the following three evaluation metrics. **Expected utility (Zintgraf et al., 2015)** ($\uparrow$). The utility function expresses the expected utility over a distribution of reward weights. **Sparsity (Xu et al., 2020)** ($\downarrow$). This metric

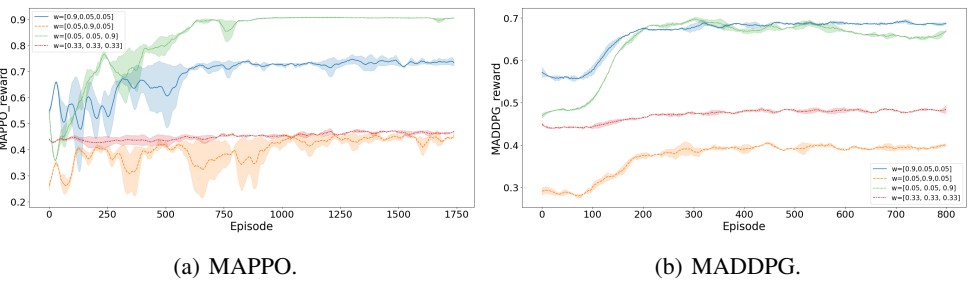

(a) MAPPO.  (b) MADDPG.

Figure 5: Reward curves of MAPPO and MADDPG.

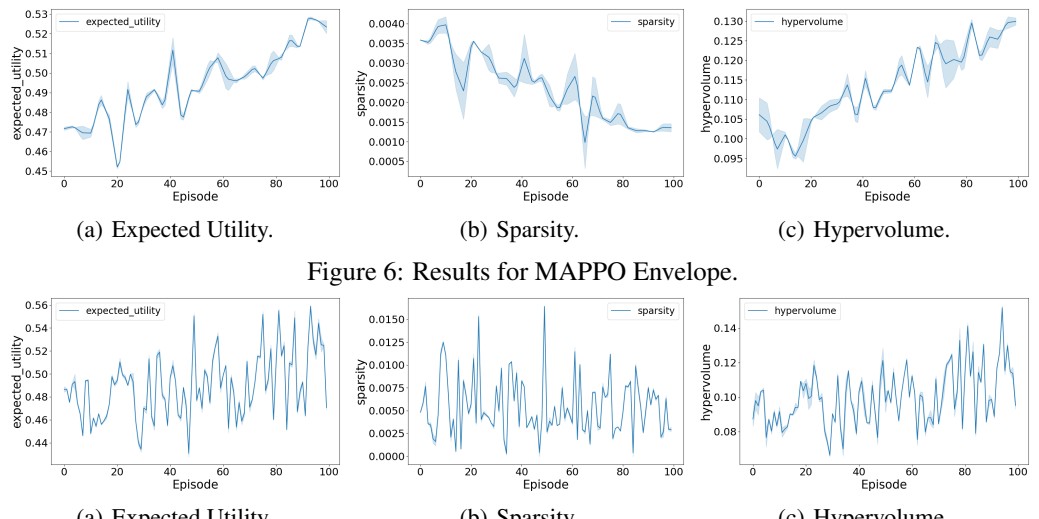

(a) Expected Utility.     (b) Sparsity.     (c) Hypervolume.

Figure 6: Results for MAPPO Envelope.

(a) Expected Utility.     (b) Sparsity.     (c) Hypervolume.

Figure 7: Results for MAPPO Prediction Guide (PG).

characterizes the diversity of the policies in a given Pareto front. **Hypervolume (Zitzler, 1999)** ($\uparrow$). Hypervolume assesses the optimizer's performance by simultaneously considering the proximity of points to the Pareto front, as well as their diversity and distribution. The hight the better. For a detailed mathematical expression of the metrics, please refer to the Appendix A.5.1.

## 4.3 BENCHMARK RESULTS

As illustrated in Figure 5, we conducted experiments using MAPPO and MADDPG under four different weight settings, with each episode consisting of 10 time steps. The algorithms converged after approximately 2000 episodes for MAPPO and 800 episodes for MADDPG. The results demonstrate that MAPPO consistently outperforms MADDPG across both weight settings. Notably, during the initial 1000 episodes, MAPPO exhibited greater oscillations in performance compared to MADDPG. We attribute this increased volatility to MAPPO's more extensive exploration of the action space, which introduces fluctuations in its performance metrics. This suggests that while MAPPO's exploratory strategy ultimately leads to superior performance, it also incurs higher variability during the early stages of learning.

In our experiments, we evaluated the performance of four multi-objective multi-agent reinforcement learning (MOMARL) methods: MAPPO Envelope, MAPPO PG, MADDPG Envelope, and MADDPG PG. Given the inherent challenges in achieving convergence in multi-agent environments, particularly when optimizing for multiple objectives, we increased the number of time steps per episode to 100. This extension was intended to provide more opportunities for learning within each episode and to better assess the convergence behavior of each method. As illustrated in Figures 6, 7, 8, and 9, the MAPPO Envelope method demonstrated the best performance among the four approaches across all evaluated metrics. Figure 10 draws the Pareto frontiers across different algorithms. Although its results were superior to the other methods, it is important to emphasize that the overall performance of the MAPPO Envelope method was still suboptimal, and significant room for improvement remains. Specifically, while MAPPO Envelope outperformed MAPPO PG, the

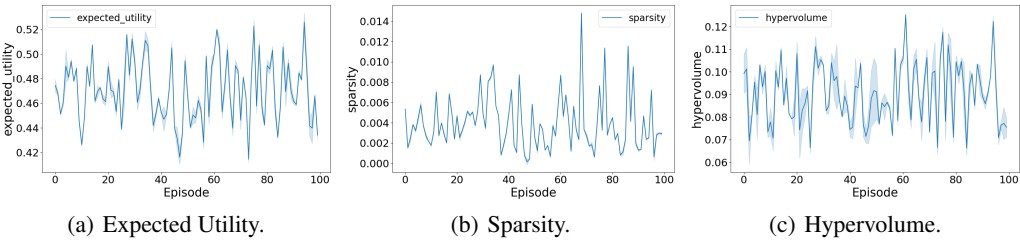

(a) Expected Utility.     (b) Sparsity.     (c) Hypervolume.

Figure 8: Results for MADDPG Envelope.

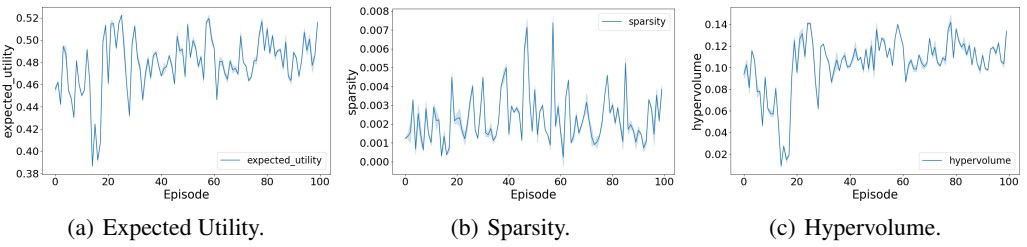

(a) Expected Utility.  (b) Sparsity.  (c) Hypervolume.

Figure 9: Results for MADDPG Prediction Guide (PG).

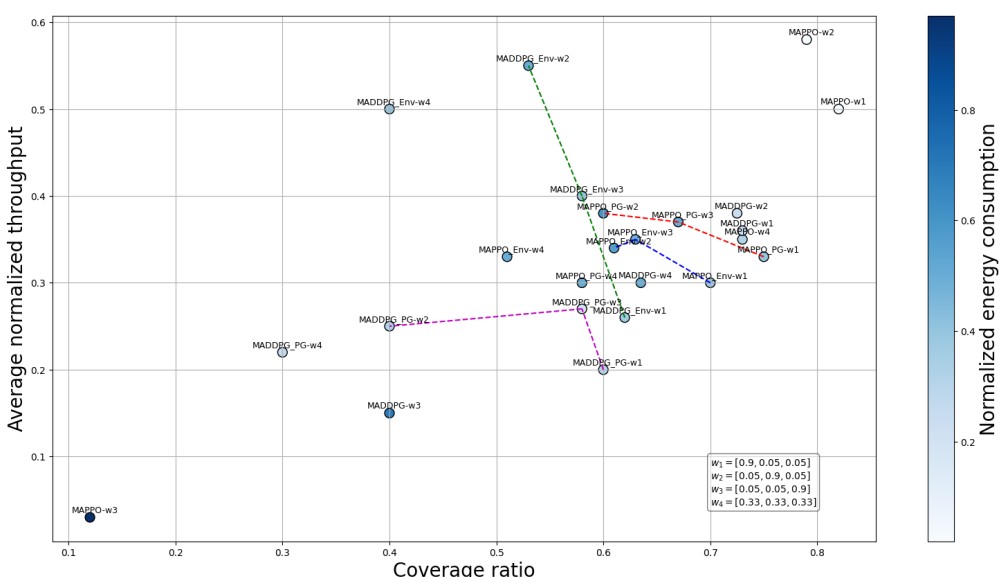

Figure 10: Pareto frontiers of different algorithms.

improvement was only marginal, and neither method achieved robust convergence. The MAPPO PG method exhibited inconsistent learning dynamics and failed to demonstrate a clear convergence pattern throughout the training process. Alternatively, the MADDPG-based methods, particularly the MADDPG Envelope, performed poorly. The MADDPG Envelope method, in particular, failed to converge altogether, reflecting its inability to handle the complexity of the multi-agent, multi-objective environment in our experiments. Similarly, the MADDPG PG method struggled with convergence, and its performance metrics lagged behind those of the MAPPO-based methods.

In conclusion, while the MAPPO Envelope method yielded the best results among the tested methods, none of the four approaches consistently achieved satisfactory convergence or optimal performance. These findings highlight the complexity and challenges inherent in multi-objective multi-agent optimization. Future research should focus on exploring more advanced methods in this area, potentially incorporating novel strategies for improving convergence and performance in MOMARL settings. Additionally, further work is needed to investigate how to more effectively balance multiple objectives across agents, which remains a critical issue in this field. Detailed results and further analysis can be found in the Appendix A.5.

## 5 CONCLUSION

In this work, we propose GenNet, a generative AI-driven mobile network simulator. To the best of our knowledge, GenNet is the first open-source, Python-based mobile network simulator equipped with a tailor-made API for reinforcement learning environments. This feature allows researchers to adjust network settings and finely evaluate various optimization strategies. Extensive experiments using multi-objective optimization algorithms in mobile networks focus on enhancing network coverage, throughput, and energy consumption, showcasing GenNet's effectiveness as a robust platform for advancing network optimization techniques.

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

## A  APPENDIX

In the appendix, we provide a comprehensive overview of the reinforcement learning interfaces used to optimize base station parameters in our wireless network environment. The RL interface plays a critical role in enabling intelligent decision-making for parameter optimization, ensuring improved coverage, throughput, and energy efficiency.

### A.1  TECHNICAL DESCRIPTIONS FOR THE MODULES USED IN GENNET

Below are detailed descriptions of the three major generative methods in our simulator GenNet.

**ActSTD** (Yuan et al., 2022): ActSTD simulates user movements between points of interest using a Generative Adversarial Imitation Learning framework. This method captures complex spatiotemporal dynamics by modeling activity trajectories as point processes. To handle irregularly sampled activities, ActSTD leverages neural differential equations to generate activities sequentially based on learned dynamics.

**MSH-GAN** (Li et al., 2024): MSH-GAN simulates mobile users' traffic usage with generative adversarial networks. It models micro-scale behavior patterns using BiLSTM networks and self-attention mechanisms, bridging these patterns to macro-scale user clusters through switch mode generators. MSH-GAN also captures multi-scale temporal dynamics with Temporal Convolutional Networks (TCNs).

**PEFNet** (Jiang et al., 2024): PEFNet simulates wireless propagation in our simulator. It combines knowledge-driven and data-driven approaches for path loss estimation in wireless transmission. PEFNet integrates Computational Electromagnetic (CEM) methods with neural networks to enhance realism, efficiency, and generalization, making it the first approach to merge these methodologies for this task.

During the training process for ActSTD, we collaborated with a major mobile network operator in China, which provided us with a dataset comprising 10,000 users in Beijing over the course of one month. This dataset includes the following categories of points of interest: Company, Concerts, Culture and Art, Education, Entertainment, Food, Government, Life Services, Market, Medicine, School, Shops, Sports, Travel, and University. The dataset records anonymous user IDs, timestamps, POI types, and latitude-longitude coordinates.

For training MSH-GAN, we again partnered with the operator to collect traffic usage data from 6,055 users over one week, specifically from April 20th to April 26th, 2016, in Shanghai. Traffic usage records were collected every half hour, with each record containing the anonymous user ID, timestamp, and traffic volume.

To train PEFNet, we utilized an open-access dataset called RSRPSet, measured by Huawei Technologies Co. This dataset contains information on base stations and receivers, as well as the corresponding Reference Signal Received Power (RSRP). There are approximately 5 million entries in the dataset, with each entry comprising 17 features. Importantly, no sensitive user information is included.

### A.2  NETWORK PERFORMANCE INDICATORS IN GENNET

Network performance encompasses a variety of metrics relevant to mobile devices, base stations, and the overall system.

**Mobile Device Indicators:** These include the device's connection status with the base station, allocated resource blocks, received power, interference power, signal-to-interference-plus-noise ratio (SINR), and communication rates.

**Base Station Performance:** Key metrics here cover the data rates provided to and required by mobile users, the number of connected users, and the classification of users into those with satisfied and unsatisfied communication demands. Additionally, this includes the measurement of energy consumption at the base station.

**System Performance:** This aspect integrates data from all users and base stations across the network. It monitors total and active users, classifying them based on whether their serviced data rates meet their desired rates. It also calculates the total of all users' desired and currently serviced data rates.

Moreover, the system tracks the coverage ratio, which indicates the proportion of the area with adequate network signal and the overall energy consumption.

## A.3 REINFORCEMENT LEARNING FRAMEWORK

We developed our RL interface using the Gym framework, which is widely recognized for its simplicity and effectiveness in designing and evaluating RL algorithms. Our environment simulates a wireless network scenario where agents (base stations) interact with the environment (network users and physical terrain) to optimize key performance metrics.

### A.3.1 ENVIRONMENT SETUP

The environment setup includes several key components:

- **State Space**: The state space represents the current configuration of the wireless network, including base station positions, antenna angles, and user distribution.
- **Action Space**: The action space defines the possible adjustments that can be made to the base station parameters, such as changing the transmission power or adjusting the antenna downtilt angle.
- **Reward Function**: The reward function provides feedback to the RL agent based on the performance of the network, with rewards for improved coverage, higher throughput, and lower energy consumption.

### A.3.2 STATE SPACE REPRESENTATION

The state space in our environment is represented by a multi-dimensional vector that includes:

- **Base Station Parameters**: Transmission power, azimuth angle, downtilt angle, horizontal beamwidth, and vertical beamwidth.
- **Network Performance Metrics**: Current coverage area, throughput, and energy consumption.
- **User Distribution**: Location and density of users within the coverage area.

This comprehensive representation ensures that the RL agent has access to all relevant information needed to make informed decisions.

### A.3.3 ACTION SPACE DESIGN

The action space is designed to allow the RL agent to make fine-grained adjustments to the base station parameters. Actions are represented as discrete or continuous changes in the following parameters:

- **Transmission Power**: Adjusting the power output of the base station to balance coverage and energy consumption.
- **Antenna Azimuth Angle**: Rotating the antenna to optimize the direction of the signal beam.
- **Antenna Downtilt Angle**: Adjusting the angle of the antenna to control the vertical spread of the signal.
- **Horizontal Beamwidth**: Modifying the width of the signal beam in the horizontal plane.
- **Vertical Beamwidth**: Modifying the width of the signal beam in the vertical plane.

### A.3.4 REWARD FUNCTION FORMULATION

The reward function is a critical component of the RL interface, guiding the agent towards optimal configurations. Our reward function is designed to balance multiple objectives:

$$R = \alpha \times \text{Coverage} + \beta \times \text{Throughput} - \gamma \times \text{Energy Consumption} \tag{8}$$

where $\alpha$, $\beta$, and $\gamma$ are weighting factors that determine the relative importance of each metric. This formulation ensures that the agent receives positive rewards for increasing coverage and throughput while minimizing energy consumption. Weights can be set by the user.

### A.3.5 TRAINING THE RL AGENT

Training the RL agent involves iteratively interacting with the environment to learn the optimal policy for parameter adjustments. The training process includes:

- **Exploration**: The agent explores different actions to understand their impact on network performance.
- **Exploitation**: The agent leverages its knowledge to choose actions that maximize the reward based on past experiences.
- **Policy Update**: The agent updates its policy using reinforcement learning algorithms.

## A.4 USAGE OF THE RL ENVIRONMENT

In this subsection, we provide a detailed guide on how to utilize the reinforcement learning environment developed for optimizing base station parameters. The RL environment is designed to simulate a wireless network, allowing for the optimization of key performance metrics such as coverage, throughput, and energy consumption.

### A.4.1 ENVIRONMENT INITIALIZATION

To begin using the RL environment, it is essential to initialize it correctly. Below are the steps and code examples to set up the environment.

**Import Necessary Libraries**   First, import the necessary libraries and modules required for the environment.

```
import pycomm  # The client interface of the simulator
from pycomm import CoverageEnv
```

**Create and Reset Environment**   Create an instance of the environment and reset it to start the simulation.

```
# Create the RL environment
env = CoverageEnv(job="coverage", port=51410)

# Reset the environment to get the initial observation
obs, info = env.reset()
```

### A.4.2 ACTION SPACE AND OBSERVATION SPACE

Understanding the action space and observation space is crucial for interacting with the environment effectively.

**Get Action and Observation Space**   Retrieve and print the action space and observation space to understand their formats and value ranges.

```
# Get action space
action_space = env.action_space
print("Action Space:", action_space)

# Get observation space
observation_space = env.observation_space
print("Observation Space:", observation_space)
```

**Action Space Details**   The action space includes the possible actions that the agent can take, such as adjusting the base station's transmission power, azimuth angle, downtilt angle, horizontal beamwidth, and vertical beamwidth. Each action is represented as a discrete or continuous value within a specified range.

**Observation Space Details** The observation space provides the state of the environment, including the current configuration of the base station parameters, network performance metrics (coverage, throughput, energy consumption), and user distribution. The observation space is represented as a multi-dimensional vector.

### A.4.3 ENVIRONMENT STEP FUNCTION

To interact with the environment, the agent performs actions, and the environment responds with new observations, rewards, and additional information.

**Step Function Execution** The step function is used to execute an action in the environment and obtain the resulting state, reward, and other information.

```
# Sample a random action
action = env.action_space.sample()

# Perform the action and get the results
next_state, reward, done, info = env.step(action)

# Print the results
print("Next State:", next_state)
print("Reward:", reward)
print("Done:", done)
print("Info:", info)
```

**State and Reward Information** The state information includes the updated base station parameters and network performance metrics. The reward is calculated based on the improvement in coverage, throughput, and energy efficiency.

### A.4.4 TASKS AND CONSTRAINTS

The environment supports various optimization tasks and constraints to simulate real-world scenarios.

**Supported Tasks** The primary tasks supported by the environment include:

- **Coverage Optimization**: Maximizing the area covered by the base station signal.
- **Throughput Optimization**: Enhancing the data transmission rate across the network.
- **Energy Efficiency Optimization**: Reducing the energy consumption of base stations while maintaining performance.

**Supported Constraints** The environment imposes constraints on the actions to ensure realistic and feasible configurations:

- **Power Constraints**: Limits on the transmission power of base stations.
- **Antenna Angle Constraints**: Restrictions on the azimuth and downtilt angles of antennas.
- **Beamwidth Constraints**: Constraints on the horizontal and vertical beamwidth of the signal.

### A.4.5 VISUALIZATION SUPPORT

The environment includes visualization capabilities to render the network and base station configurations, providing insights into the optimization process.

**Render Environment** The following code snippet demonstrates how to render the environment.

```
# Render the environment
env.render()
```

The rendering provides a visual representation of the base stations, coverage areas, and user distribution, aiding in the analysis and debugging of the optimization process.

### A.4.6 PERFORMANCE METRICS

To evaluate the performance of the RL agent, we measure several key metrics:

- **Coverage Ratio**: The ratio of the area covered by the base station signals.
- **Network Throughput**: The aggregate data transmission rate achieved across the network.
- **Energy Consumption**: The total energy consumed by the base stations.

### A.4.7 EXAMPLE USAGE

Below is a complete example of initializing the environment, performing actions, and visualizing the results.

```
import pycomm
from pycomm import CoverageEnv

# Create the RL environment
env = CoverageEnv(job="coverage", port=51410)

# Reset the environment
obs, info = env.reset()

# Loop to perform actions
for _ in range(100):
    # Sample a random action
    action = env.action_space.sample()

    # Perform the action
    next_state, reward, done, info = env.step(action)

    # Render the environment
    env.render()

    # Break the loop if done
    if done:
        break

# Close the environment
env.close()
```

This example demonstrates the typical workflow for using the RL environment, from initialization to execution and visualization. By following these steps, researchers can effectively leverage the environment to optimize wireless network parameters and achieve significant improvements in performance metrics.

### A.5 ALGORITHM COMPARISON

In order to verify the functionality of our simulation platform, we compared several RL algorithms:

**MAPPO** (Yu et al., 2022): MAPPO extends the PPO algorithm for multi-agent environments, coordinating multiple agents to learn optimal policies while preventing interference among them.

**MADDPG** (Kaur et al., 2023): A multi-agent version of DDPG that uses centralized training and decentralized execution to learn policies in multi-agent environments.

**MAPPO Envelop**: Extending Envelop Yang et al. (2019) to multi-agent case with MAPPO framework to cope with large-scale network optimization problem. The Envelope algorithm manages trade-

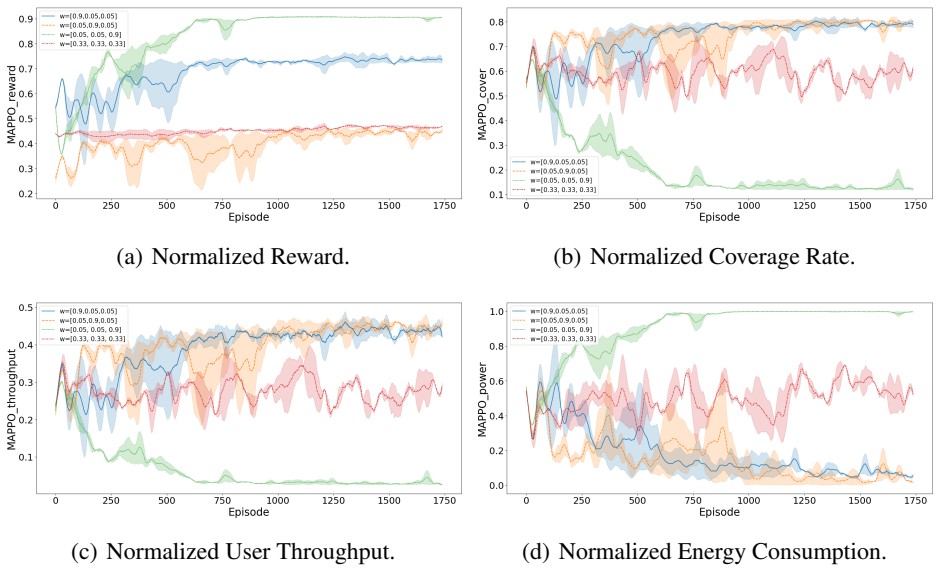

(a) Normalized Reward.

(b) Normalized Coverage Rate.

(c) Normalized User Throughput.

(d) Normalized Energy Consumption.

Figure 11: The convergence curves of MAPPO.

offs in multi-objective reinforcement learning by constructing an "envelope" around the objectives, optimizing them simultaneously without excessive bias.

**MAPPO PG**: Extending PG Xu et al. (2020) to multi-agent case with MAPPO framework to cope with large-scale network optimization problem. PG method uses predictive models to guide the RL process by predicting outcomes across multiple objectives, aiding in decision-making, and optimizing all objectives concurrently, particularly in continuous control tasks.

**MADDPG Envelop**: Extending Envelop Yang et al. (2019) to multi-agent case with MADDPG framework to cope with large-scale network optimization problem. The Envelope algorithm manages trade-offs in multi-objective reinforcement learning by constructing an "envelope" around the objectives, optimizing them simultaneously without excessive bias.

**MADDPG PG**: Extending PG Xu et al. (2020) to multi-agent case with MADDPG framework to cope with large-scale network optimization problem. PG method uses predictive models to guide the RL process by predicting outcomes across multiple objectives, aiding in decision-making, and optimizing all objectives concurrently, particularly in continuous control tasks.

### A.5.1 EVALUATION METRICS

**Expected utility (Zintgraf et al., 2015)** ($\uparrow$). The utility function expresses the expected utility over a distribution of reward weights, $\mathcal{W}$. Let $\Pi$ be a set of policies and $\tilde{\mathcal{F}} = \{v^\pi \mid \pi \in \Pi\}$ represent its corresponding approximate Pareto front. Expected utility metric is then defined as follows:

$$\text{EU}(\tilde{\mathcal{F}}) = \mathbb{E}_{\mathbf{w} \sim \mathcal{W}} \left[ \max_{\mathbf{v}^\pi \in \mathcal{F}} \mathbf{v}^\pi \cdot \mathbf{w} \right]. \tag{9}$$

**Sparsity (Xu et al., 2020)** ($\downarrow$). This metric characterizes the diversity of the policies in a given Pareto front, which is given by:

$$\text{S}(\tilde{\mathcal{F}}) = \frac{1}{|\tilde{\mathcal{F}}| - 1} \sum_{j=1}^{m} \sum_{i=1}^{|\tilde{\mathcal{F}}|-1} \left( \mathcal{L}_j(i) - \mathcal{L}_j(i+1) \right)^2, \tag{10}$$

where $\mathcal{L}_j$ is the sorted list of the values of the $j$-th objective considering all policies in $\tilde{\mathcal{F}}$, and $\mathcal{L}_j(i)$ is the $i$-th value in $\mathcal{L}_j$.

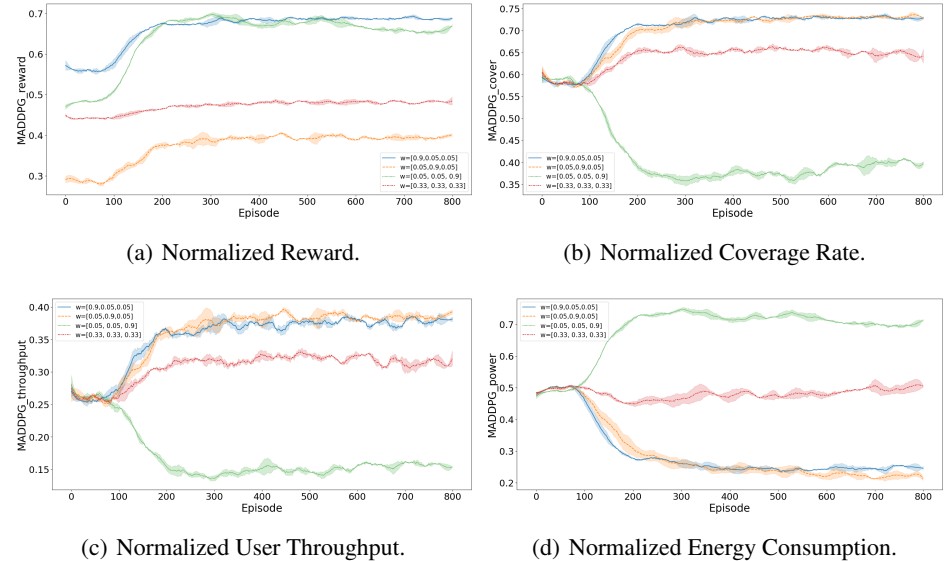

(a) Normalized Reward.

(b) Normalized Coverage Rate.

(c) Normalized User Throughput.

(d) Normalized Energy Consumption.

Figure 12: The convergence curves of MADDPG.

**Hypervolume (Zitzler, 1999)** ($\uparrow$). Given a reference point, $v_{\text{ref}}$, the hypervolume metric is defined as:

$$\text{HV}\left(\tilde{\mathcal{F}}, \mathbf{v}_{\text{ref}}\right) = \bigcup_{\mathbf{v}^\pi \in \tilde{\mathcal{F}}} \text{volume}\left(\mathbf{v}_{\text{ref}}, \mathbf{v}^\pi\right), \tag{11}$$

where volume $\left(v_{\text{ref}}, v^\pi\right)$ is the volume of the hypercube spanned by the reference vector, $v_{\text{ref}}$, and the vector, $v^\pi$.

### A.5.2 ANALYSIS OF MAPPO

- **(a) reward**:
    - Different weights (w) show varying trends in the reward metric.
    - Weight $w = [0.05, 0.05, 0.9]$ (green) achieves the highest reward, indicating that this weight configuration leads to the best expected reward performance.
    - Other weight configurations have relatively lower and stable rewards.
- **(b) cover**:
    - In the cover metric, weight $w = [0.05, 0.05, 0.9]$ (green) performs poorly, nearly at 0. Weight $w = [0.05, 0.9, 0.05]$ (orange) performs best.
    - Other weight configurations achieve relatively higher and more fluctuating cover values.
- **(c) throughput**:
    - In the throughput metric, weight $w = [0.05, 0.05, 0.9]$ (green) also performs poorly, nearly at 0.
    - Other weight configurations maintain higher and relatively stable throughput.
- **(d) power**:
    - In the power metric, weight $w = [0.05, 0.05, 0.9]$ (green) reaches a level close to 1.
    - Other weight configurations have lower and stable power values.

### A.5.3 ANALYSIS OF MADDPG

- **(a) Reward**:
    - Weight $w = [0.9, 0.05, 0.05]$ (blue) achieves the highest reward, indicating that this weight configuration leads to the best expected reward performance.
    - Other weight configurations have relatively lower and stable rewards.
- **(b) Cover**:

- In the cover metric, weight $w = [0.05, 0.05, 0.9]$ (green) performs poorly, with a rapid decline.
- Other weight configurations achieve higher and more stable cover values.

- **(c) Throughput**:
  - In the throughput metric, weight $w = [0.9, 0.05, 0.05]$ (green) performs poorly. Weight $w = [0.05, 0.9, 0.05]$ (orange) performs best.
  - Other weight configurations maintain higher and relatively stable throughput.

- **(d) Power**:
  - In the power metric, weight $w = [0.05, 0.05, 0.9]$ (green) reaches the highest level initially but declines steadily.
  - Other weight configurations have lower and more stable power values.

### A.5.4 COMPARISON OF MAPPO AND MADDPG

- **Reward**:
  - MAPPO achieved a higher reward value than MADDPG.
  - MADDPG shows faster convergence in reward compared to MAPPO.
- **Cover**:
  - MAPPO with weight $w = [0.05, 0.05, 0.9]$ performs poorly in cover, while MADDPG shows a rapid decline initially.
  - Other weight configurations in both algorithms achieve higher cover values, with MADDPG generally showing more stability.
- **Throughput**:
  - Both MAPPO and MADDPG perform poorly in throughput with weight $w = [0.05, 0.9, 0.05]$.
  - Other weight configurations maintain higher throughput values, with MADDPG showing slightly better stability.
- **Power**:
  - In power, MAPPO with weight $w = [0.05, 0.05, 0.9]$ reaches close to 1, while MADDPG initially reaches a high level but declines.
  - Other configurations have lower power values, with MADDPG showing more steady performance over time.

From the above analysis, we can see that under different preference weights, the single-objective multi-agent algorithm can learn according to the pre-set preferences. The experimental results verify the correctness of our simulator.

### A.5.5 ANALYSIS OF PARETO FRONTS

As shown in Figure 10, we tested the four preferences set in MAPPO and MADDPG by inputting them into the multi-objective reinforcement learning algorithms and tried to draw their Pareto frontiers.

- **Pareto Fronts for MAPPO**:
  - The Pareto front for MAPPO includes points such as MAPPO-w1 and MAPPO-w2, which exhibit high average normalized throughput and high coverage ratio.
  - MAPPO-w3 shows significantly lower values in both metrics, indicating it is not part of the Pareto optimal solutions.

- **Pareto Fronts for MADDPG**:
  - The Pareto front for MADDPG includes points such as MADDPG-Env-w2 and MADDPG-Env-w3, which have balanced performance in terms of average normalized throughput and coverage ratio.
  - MADDPG-w3, on the other hand, shows poor performance in both metrics, making it a non-optimal solution.

- **Comparison of MAPPO and MADDPG**:
  - MAPPO-w1 and MAPPO-w2 achieve higher normalized energy consumption but perform well in throughput and coverage ratio, indicating a trade-off between energy consumption and performance.
  - MADDPG generally shows more balanced performance across different metrics, with multiple points lying close to the Pareto front.
  - MAPPO demonstrates higher variability in performance based on different weight configurations compared to MADDPG.

### A.5.6 REASONS FOR NON-CONVERGENCE IN MULTI-OBJECTIVE REINFORCEMENT LEARNING (MORL)

- **Conflicting Objectives**:
  - In our multi-object problem, objectives can conflict, making it difficult for the agent to optimize all objectives simultaneously.
  - This conflict can lead to oscillations in policy updates, preventing convergence.

- **High Dimensionality**:
  - The state and action spaces in wireless optimization multi-object problem are high-dimensional, increasing the complexity of the learning problem.
  - High-dimensional spaces require more data and computation to explore effectively, which can slow down the learning process and hinder convergence.

- **Credit Assignment Problem**:
  - Assigning credit for outcomes to specific actions becomes more challenging with multiple objectives, especially when the effects of actions are delayed.
  - This problem can lead to suboptimal policy updates, preventing the agent from learning the optimal policy.

- **Exploration vs. Exploitation Dilemma**:
  - Balancing exploration of the environment and exploitation of known rewards is more complex with multiple objectives.
  - Insufficient exploration can lead to premature convergence to suboptimal policies, while excessive exploration can slow down convergence.

- **Algorithmic Limitations**:
  - Current MORL algorithms may have inherent limitations in handling the trade-offs between multiple objectives effectively.
  - Improvements in algorithm design and the incorporation of domain knowledge are necessary to address these limitations.

