# OpenReview forum: "GenNet: A Generative AI-Driven Mobile Network Simulator for Multi-Objective Network Optimization"
_ICLR.cc/2025/Conference — Submitted to ICLR 2025_

### Official Review · Reviewer_yAjY · 2024-11-01

**Soundness:** 2
**Presentation:** 3
**Contribution:** 3
**Rating:** 6
**Confidence:** 4

**Summary:**

This paper develops GenNet, a generative AI-driven mobile network simulator designed for mulit-objective network optimization with reinforcement learning. GenNet can create virtual replicas of mobile users, base stations, and wireless environments, utilizing generative AI methods to simulate the behaviors of these entities under various network settings with high accuracy. It utilizes generative AI methods to accurately simulate the behaviors of these entities under various network settings based on extensive real-world data, thereby surpassing traditional discrete-event-driven simulations.

**Strengths:**

1. This paper proposes the first open-source generative AI-driven mobile network simulator, which breaks away from traditional event-driven simulators and shows originality by leveraging data-driven generative AI to enhance realism and simulation efficiency.
2. GenNet’s compatibility with RL-based optimizations and multi-objective network tuning positions it as a potentially impactful solution for researchers and practitioners in mobile networking.
3. The paper is well-structured, with a clear logical flow that guides readers through the motivations, design, implementation, and evaluation of GenNet. The clarity in writing enhances the paper's readability.

**Weaknesses:**

1. Ideally, Table 1 should be one of the most critical tables in the paper, as it is intended to reveal the advantages of GenNet over other simulators. However, the table does not make GenNet’s superiority immediately apparent, aside from its support for realistic traffic. Furthermore, the paper itself acknowledges GenNet’s lack of a comprehensive protocol stack, which raises further questions about its value compared to existing simulators.
2. Although the simulation results are promising, the paper lacks an evaluation of GenNet’s performance in actual network environments or a discussion on the challenges and potential adjustments needed for deployment in real-world network conditions.
3. The simulators selected for comparison with GenNet, such as CityFlow and MATSim, are relatively dated, which may not fully reflect the current advancements in network simulation technology. To strengthen the evaluation, the authors are encouraged to include comparisons with more recent simulators.

**Questions:**

1. It seems that the AI-driven simulator proposed primarily involves the straightforward application of generative models like GANs, VAEs, and diffusion models. Please provide specific details on any modifications or innovations they made to the generative models to adapt them for mobile network simulation.
2. The introduction does not clearly outline any specific challenges in implementing this AI-driven simulator. With available datasets and pre-existing AI models, it is unclear what technical or conceptual work was required to make this solution viable. Please add a specific section in the introduction or methodology detailing the key technical challenges in developing GenNet and how you overcame them.
3. In Table 2, the primary advantage of GenNet appears to be reduced computation time. However, given that this is a simulation platform for validating optimization algorithms, it is unclear whether computation time is a critical factor. Please provide specific examples or use cases where reduced computation time in network simulations leads to tangible benefits in research or industry applications. This would help justify the importance of GenNet's speed improvements.
4. For the experimental results in Figures 7, 8, and 9, how should these be interpreted? What are the evaluation criteria for each metric, and how do these criteria indicate that MAPPO PG is optimal?  Please add detailed explanations of each metric directly in the figure captions or in an accompanying text box. Additionally, please provide a clear interpretation guide for these figures in the main text, explaining how to read the graphs and what conclusions can be drawn from them.
5. Which specific datasets were used for training GenNet? Could this choice potentially limit the system’s generalizability or make it overly dependent on particular datasets? Please provide a detailed description of the datasets used, including their size, diversity, and any potential biases. Additionally, please discuss how you addressed or mitigated potential overfitting to these specific datasets, and what steps you took to ensure GenNet's applicability to a wide range of network scenarios.

---

> ### Author Response · Authors · 2024-11-26
>
> We thank the reviewer for recognizing our paper’s strengths and giving constructive comments. We would like to address the comments and questions below.
>
> **Q1**: Please provide specific details on any modifications or innovations they made to the generative models to adapt them for mobile network simulation.
>
> **A1**: The generative models used in the proposed simulator have undergone specific modifications to better suit mobile network environments. For example, the user mobility model leverages neural differential equations within a generative adversarial imitation learning framework to capture continuous spatiotemporal dynamics, addressing the irregularity and sparsity of user activity patterns. This allows for realistic simulation of individual-level trajectories, important for mobile scenarios. Additionally, for wireless propagation, PEFNet, a physics-informed neural network, integrates Volume Integral Equations (VIE) from Maxwell's equations to model electromagnetic signal propagation with high physical fidelity. PEFNet also uses a data-driven learning component to refine path loss estimates with real-world data, ensuring accuracy and adaptability across different urban environments. These innovations collectively enhance the accuracy and utility of the simulator in accurately representing mobile network dynamics.
>
> **Q2**: Please add a specific section in the introduction or methodology detailing the key technical challenges in developing GenNet and how you overcame them.
>
> **A2**: The key technical challenges in developing GenNet included:
>
> 1. Scalable user behavior simulation: Simulating millions of mobile users realistically was a significant challenge. We used advanced generative AI techniques (GANs and VAEs) to create detailed and diverse user movement and communication patterns, ensuring high scalability and accuracy.
>
> 2. High-fidelity network component modeling: Modeling network elements like base stations and wireless environments required capturing detailed configurations and environmental factors. We used operator data combined with stochastic, deterministic and e data-driven model to achieve accurate simulations of signal propagation.
>
> 3. Efficient parallel processing: Handling real-time interactions of millions of users required efficient computation. We addressed this by implementing parallelization strategies with synchronous updates, mutual exclusion, and coroutines to allow concurrent simulation tasks, thus enhancing scalability and efficiency.
>
> These solutions allowed GenNet to overcome scalability, fidelity, and efficiency challenges, resulting in a robust digital twin for mobile network optimization.
>
> **Q3**: Please provide specific examples or use cases where reduced computation time in network simulations leads to tangible benefits in research or industry applications.
>
> **A3**: Reduced computation time in network simulations has significant reality benefits, especially in scenarios involving reinforcement learning (RL). In RL algorithms, the agent needs to interact with the simulator tens of thousands of times to learn optimal strategies. Improved simulation efficiency drastically accelerates this iterative process, leading to faster training and convergence of the RL model. This means that new network optimization policies can be developed, tested, and deployed much more quickly, providing substantial advantages in both research exploration and industrial applications where rapid iteration is important.
>
> **Q4**: For the experimental results in Figures 7, 8, and 9, how should these be interpreted? please provide a clear interpretation guide for these figures in the main text, explaining how to read the graphs and what conclusions can be drawn from them.
>
> **A4**: The experimental results in Figures 7, 8, and 9 illustrate the trends in different performance metrics during the iterative process of each corresponding algorithm. It is important to note that none of the algorithms have fully converged in these figures. These figures are intended to show how key metrics evolve over time with different algorithms. From these results, we observe that in large-scale mobile network environments, existing multi-objective reinforcement learning algorithms struggle significantly to achieve convergence, indicating the challenges of applying these methods to complex, dynamic mobile network scenarios.

---

> ### Author Response · Authors · 2024-11-26
>
> **Q5**: Which specific datasets were used for training GenNet? Could this choice potentially limit the system’s generalizability or make it overly dependent on particular datasets?
>
> **A5**: In the development of GenNet, we used several primary datasets. For example:
>
> 1. Mobile network operator dataset: This dataset includes data from a major mobile network operator in China, covering the activity trajectories of 10,000 users over one month in the Beijing area. It contains information about user activities at various locations, such as offices, restaurants, and schools, used for simulating user behavior trajectories.
>
> 2. Foursquare dataset: This dataset records activities of 1,000 users over a month, including visits to different types of locations such as arts and entertainment venues, food and drink establishments, and office spaces. It helps in simulating diverse user behavior patterns.
>
> 3. RadioMapSeer dataset: This dataset contains building layouts, base station positions, and path loss maps in the form of images. It is primarily used for training the model to understand the relationship between environmental geography and path loss.
>
> 4. RSRPSet dataset: The RSRPSet dataset, published by Huawei Technologies, includes data about base station and receiver positions, as well as Reference Signal Received Power (RSRP), with approximately 5 million entries. While specific geographic details were removed for privacy reasons, the dataset was processed to reconstruct the geographical layout for training and testing path loss estimation models.
>
> These datasets play a crucial role in providing realistic environment modeling and effective training support. However, since the data is sourced from specific geographic regions (e.g., the Beijing area and specific Foursquare locations) and application scenarios, this might limit the generalizability of GenNet, making it more reliant on certain environments. Future work could focus on expanding the range of geographic areas and user behavior data to enhance GenNet’s applicability and robustness across diverse scenarios.
>
> **Q6**: Please discuss how you addressed or mitigated potential overfitting to these specific datasets, and what steps you took to ensure GenNet's applicability to a wide range of network scenarios.
>
> **A6**: To address the potential risk of overfitting to the specific datasets used for GenNet, we implemented several key strategies to enhance the generalizability and applicability of the model across a wide range of network scenarios:
>
> 1. Diverse dataset integration: We combined multiple datasets from different sources, including the mobile network operator dataset, Foursquare data, RadioMapSeer, and RSRPSet. Each dataset has unique characteristics—ranging from user activity data in urban settings to different network path loss scenarios—ensuring that GenNet was exposed to diverse user behaviors and network conditions during training. This helped mitigate overfitting to any one specific dataset and improved the robustness of the model.
>
> 2. Cross-validation across datasets: We applied cross-validation across different datasets, such as using RadioMapSeer for training and RSRPSet for validation. This cross-validation approach ensured that GenNet was tested on environments it had not seen during training, thereby helping to assess and improve its performance across different network configurations and reduce overfitting risks.
>
> 3. Physics-informed neural networks: In path loss modeling, we incorporated a Physics-Informed Neural Network (PINN) approach by embedding electromagnetic theory into the training process. This involved using the Volume Integral Equation (VIE) as a knowledge-driven component alongside data-driven training, which helped the model to adhere to physical laws governing signal propagation. By incorporating physical knowledge, GenNet could generalize well beyond the specific patterns present in the data, leading to better applicability across various network conditions.
>
> These measures are crucial in mitigating overfitting and ensuring that GenNet could generalize effectively to a wide range of network scenarios and environments.

---

> > ### Comment · Reviewer_yAjY · 2024-11-27
> >
> > Thanks the authors for their effort in addressing my concerns.

---

### Official Review · Reviewer_9XRN · 2024-11-02

**Soundness:** 2
**Presentation:** 3
**Contribution:** 2
**Rating:** 3
**Confidence:** 5

**Summary:**

This paper develops a network simulator for multi-objective optimization using generative AI and establishes a reinforcement learning interface. Based on a grey-box approach, the paper constructs modules for user mobility, wireless environment, and models base stations and network performance. The authors employ various generative algorithms to simulate user movement, service traffic, and wireless propagation models. Based on this simulator, the authors conducted reinforcement learning experiments to optimize base station coverage, throughput, and energy efficiency.

**Strengths:**

The authors propose the use of generative AI to construct modules in communication network simulators, which is a good design idea for simplifying network simulators.

**Weaknesses:**

The simulator developed by the author claims to be fast. In reality, since the simulator is only used for specific functions such as traffic, coverage, and energy optimization, it cannot control other functions of wireless networks and cannot observe other performance metrics of wireless networks, such as slice control, congestion control, quality of service assurance, deterministic guarantees, etc. When these functions are added, the software scale and simulation speed will also increase. Moreover, due to the black-box nature of generative AI, the coexistence of AI modules with different functions is questionable. That is to say, the scalability of the system is questionable.
By searching the paper title on GitHub, the paper’s code information has been retrieved. This seems to violate the double-blind peer review agreement. By analyzing the code, it can be concluded that the work is aimed at a specific reinforcement learning algorithm, and the simulation work uses generative AI to simulate some functions.

**Questions:**

1. Due to the use of generative AI technology, the fidelity of its capabilities compared to traditional system-level simulation tools is questionable. Please explain the accuracy of the generated information.
2. The simulation experiment uses GPU and generative AI to simulate system functions, compared to traditional simulation software that uses CPU. It's unfair. In fact, there is also a large amount of work using GPU to accelerate simulation experiments. If GPU acceleration is used in both cases, what conclusion would the author draw about the performance comparison?
3. Could the authors explain how many control parameters, performance metrics, and functions need to be retained in a comprehensive simulator, such as NS-3, OPNET, OMNET++, to be generally applicable to most simulation experiments? What is the framework for the full implementation of the simulator proposed by the author, and how is its scalability ensured?
4. Please clarify the statement about MATLAB in the INTRODUCTION Section. The author mentioned in the related work that MATLAB is a link-level simulator. As we know, MATLAB provides a scientific computing tool. Based on this tool, link-level simulators and system-level simulators can be developed. For example, the well-known Vienna Platform can implement the protocol stack for LTE and 5G systems and complete system-level simulations.
5. The author claims in the introduction that the simulation speed of NS-3/OPNET/MATLAB/OMNET++ is slower than real networks. Please provide evidence or citations supporting their claim about simulation speeds.In my opion, discrete event-driven simulators do not require waiting for physical time to elapse and are faster than validating real networks.
6. In section 2.1, the author claims that OPNET/NS-3 lacks user mobility modeling and is not suitable for mobile access networks. In fact, these simulators have interfaces or are open-source, allowing for the development of mobility models and are widely used in mobility experiments. Many research works on mobility based on these simulators can be found through Google Scholar. Please clarify your statement.
7. Please provide the system architecture about your proposed scable, full functional system-level simulator.

---

> ### Author Response · Authors · 2024-11-26
>
> We thank the reviewer for recognizing our paper’s strengths and giving constructive comments. We would like to address the comments and questions below.
>
> **Q1**: Please explain the accuracy of the generated information.
>
> **A1**: The accuracy of the generated information is supported by the experimental results of the algorithms used in our simulator. For example, in the activity trajectory generation, the framework based on generative adversarial imitation learning, achieved over 50% reduction in MAPE when simulating COVID-19 spread compared to baseline models, indicating a high level of accuracy. For the path loss estimation, the PEFNet model achieved an R² of 0.9731, a MAE of 0.7759 dB, and a RMSE of 2.1733 dB on the RadioMapSeer dataset, demonstrating better accuracy compared to other baseline models. These results validate the high accuracy and reliability of the information generated by our simulator.
>
> **Q2**: The simulation experiment uses GPU and generative AI to simulate system functions, compared to traditional simulation software that uses CPU. If GPU acceleration is used in both cases, what conclusion would the author draw about the performance comparison?
>
> **A2**: Our simulator supports GPU acceleration, unlike traditional simulation software which typically lacks GPU acceleration capabilities. This fundamental difference is important for the performance comparison, as the ability to leverage GPU acceleration can significantly enhance simulation efficiency and scalability, providing a major advantage over conventional CPU-based simulators.
>
> **Q3**: Could the authors explain how many control parameters, performance metrics, and functions need to be retained in a comprehensive simulator, such as NS-3, OPNET, OMNET++, to be generally applicable to most simulation experiments?
>
> **A3**: To address the question regarding the number of control parameters, performance metrics, and functions in a comprehensive simulator like NS-3, OPNET, or OMNeT++, it should be noted that these numbers depend on the network type and use case. However, a generally versatile simulator should include around 20 to 30 control parameters covering protocol settings, node mobility, channel, and link characteristics. It should also offer 10 to 15 key performance metrics, such as throughput, delay, jitter, packet delivery ratio, and energy consumption. Additionally, hundreds of functions are needed to model various network protocols, routing algorithms, and traffic generation features, ensuring the simulator’s adaptability to a wide range of network scenarios.
>
> **Q4**: Please clarify the statement about MATLAB in the INTRODUCTION Section. The author mentioned in the related work that MATLAB is a link-level simulator. As we know, MATLAB provides a scientific computing tool. Based on this tool, link-level simulators and system-level simulators can be developed. For example, the well-known Vienna Platform can implement the protocol stack for LTE and 5G systems and complete system-level simulations.
>
> **A4**: MATLAB itself is not a simulator for system-level network simulations; rather, it is a scientific computing tool that excels at numerical analysis and matrix computations. Its capabilities are more suited for link-level analysis, where individual connections and their behaviors can be modeled in detail. The Vienna platform is indeed developed based on MATLAB, but it extends beyond MATLAB’s standard capabilities by incorporating additional modules and complex implementations to support both link-level and system-level simulations for technologies like LTE and 5G. Therefore, while MATLAB provides the foundation for link-level analysis, it lacks the built-in functionalities for comprehensive system-level simulations without significant extensions like those provided by the Vienna Platform.
>
> **Q5**: The author claims in the introduction that the simulation speed of NS-3/OPNET/MATLAB/OMNET++ is slower than real networks. Please provide evidence or citations supporting their claim about simulation speeds.
>
> **A5**: The statement in the introduction regarding the simulation speed of NS-3, OPNET, MATLAB, and OMNeT++ being slower than real networks is supported  by [1]. These simulators often execute 3-4 orders of magnitude slower than real-time networks due to the intricacies involved in modeling packet-level processes, such as congestion control and queueing mechanisms, under pre-defined simulation rules. This nature although essential for accurate emulation, leads to significant computational overhead, making them inefficient for real-time or large-scale simulations.
>
> [1] Linbo Hui, Mowei Wang, Liang Zhang, Lu Lu, and Yong Cui. Digital twin for networking: A data-driven performance modeling perspective. IEEE Network, 2022.

---

> > ### Author Response · Authors · 2024-11-26
> >
> > **Q6**: In section 2.1, the author claims that OPNET/NS-3 lacks user mobility modeling and is not suitable for mobile access networks. In fact, these simulators have interfaces or are open-source, allowing for the development of mobility models and are widely used in mobility experiments. Please clarify your statement.
> >
> > **A6**: The statement regarding OPNET and NS-3 lacking mobility modeling capabilities refers to the fact that these simulators do not have built-in user mobility modeling features. While it is true that they are open-source or provide interfaces that allow the integration of external mobility models, the implementation of such models typically relies on additional libraries or third-party extensions. The core functionality of OPNET/NS-3 does not directly include user mobility modeling, and the support for mobile access networks requires significant customization or integration of external tools to achieve this capability.
> >
> > **Q7**: Please provide the system architecture about your proposed scable, full functional system-level simulator.
> >
> > **A7**: The system architecture can be described in three main components: mobile user modeling, base station modeling, and wireless environment modeling. Firstly, the mobile user modeling module uses Generative Adversarial Networks (GANs) and multi-scale generation models to simulate user mobility and communication behaviors, covering different user types such as vehicles and pedestrians. These virtual users provide detailed representations of real-world behaviors during simulation. Secondly, the base station modeling module precisely replicates base station physical attributes such as geographic location, transmission power, and antenna configuration, enabling flexible parameter adjustments to simulate the impact of different settings on network performance. Lastly, the wireless environment modeling module combines stochastic and deterministic models to simulate the impact of physical factors, like buildings and terrain, on signal propagation, enhancing the accuracy and applicability of signal modeling. These three components are integrated to form a complete digital twin platform for mobile networks.
> >
> > The system operates as a data flow process, beginning with input data and simulation results managed through MongoDB and PostgreSQL databases. Users can submit simulation requests through the client using gRPC or HTTP protocols, including control parameters and job names. Upon receiving the request, the system initializes the simulation environment, conducting an integrated simulation of mobile users, base stations, and the wireless environment. The simulation process leverages parallel processing to enhance efficiency, allowing multiple simulation tasks to run concurrently without interference. Once the simulation is complete, the system stores the results in the database and sends a response to the client containing reward information and the simulation status. This design allows the system to achieve accurate large-scale user behavior simulations while providing a real-time, efficient solution for network optimization and management.

---

> > ### Comment · Reviewer_9XRN · 2024-11-27
> >
> > I appreciate the authors for their efforts in addressing my concerns. However, I still have some questions to the answers.
> > To A1: The reviewer hopes the authors can explain whether the GenAI-based simulator proposed by the authors has better or worse accuracy compared to traditional system-level simulation platforms based on system implementation, and the reasons for this.
> > To A2: In fact, some simulation software has recently started using GPUs for acceleration. Numerous papers have discussed the work on accelerating traditional simulation systems with GPUs.
> > To A3: Since system-level simulation platforms have a large number of control optimization parameters and key performance indicators, they can conduct a wide range of simulation experiments. Please explain whether the GenAI-based solution proposed by the authors, if it achieves similar functionality, still maintains advantages in terms of software size and simulation speed.
> > To A4: Matlab is neither a system-level nor a link-level simulation platform; it is a scientific computing tool. In my view, comparing it to a simulation platform is inappropriate.
> > To A5: System-level simulation platforms use a discrete-time-driven approach for computation. During experiments, they do not require real-time waiting or the processing of actual data packets. Consequently, simulation platforms typically operate faster than real networks. This is also why many experiments are conducted in DTN environments rather than on real networks. The literature provided by the authors states that the implementation of DTN, unlike simulators, can forgo complex mechanisms, allowing it to operate faster than real networks. This contradicts the authors' claim that simulators are slower than real networks.

---

> > > ### Author Response · Authors · 2024-12-01
> > >
> > > **Q8**: The reviewer hopes the authors can explain whether the GenAI-based simulator proposed by the authors has better or worse accuracy compared to traditional system-level simulation platforms based on system implementation, and the reasons for this.
> > >
> > > **A8**: The accuracy of our proposed GenAI-based simulator has been addressed in our response to Question 1. To clarify, our approach achieves higher accuracy compared to methods used in traditional system-level simulation platforms, such as 3GPP TR 36.873, 3GPP TR 38.901, and Raytracing (a widely-used deterministic channel model). This improved accuracy is demonstrated in the paper [1]. The main advantage of our approach is that Physics-informed Neural Networks (PINNs) integrate physical laws directly into the training process, enabling more precise path loss estimation. Unlike traditional models, which rely on empirical data and simplifications, PINNs leverage both physical principles and limited data to achieve better accuracy, generalization, and efficiency.
> > >
> > > [1] Fenyu Jiang, Tong Li, Xingzai Lv, Hua Rui, and Depeng Jin, "Physics-informed neural networks for path loss estimation by solving electromagnetic integral equations," IEEE Transactions on Wireless Communications, 2024.
> > >
> > > **Q9**: In fact, some simulation software has recently started using GPUs for acceleration. Numerous papers have discussed the work on accelerating traditional simulation systems with GPUs.
> > >
> > > **A9**: It is true that some papers have explored the use of GPUs to accelerate traditional simulation systems. However, our contribution lies in the development of a new simulator based on generative AI, rather than improving existing traditional simulation systems. For a fair comparison, we only benchmarked against the original versions of these systems. We want to emphasize that our approach shifts from the traditional event-triggered simulation method to a data generation-based simulation framework, enabling natural GPU acceleration without the need for external patches or modifications. This fundamental change allows us to fully leverage GPU capabilities for efficient and scalable simulations.
> > >
> > > Unlike traditional systems that primarily focus on speeding up the computation of predefined models (e.g., 3GPP, Raytracing), GenNet incorporates an advanced simulation engine that integrates with external optimizers via custom RPCs. This architecture not only supports GPU-based acceleration for computational efficiency but also enables real-time network optimization and dynamic interaction handling through parallelization and coroutine management, ensuring that complex interactions between mobile users and base stations are simulated efficiently.
> > >
> > > Moreover, our approach eliminates the need for large amounts of empirical data by combining physical law-based modeling with real-time network interactions, providing both computational speed and accuracy. This makes GenNet not only faster but also more flexible and accurate in simulating complex, real-world scenarios, compared to traditional GPU-accelerated systems that rely on simpler or fixed models.

---

> > > > ### Author Response · Authors · 2024-12-01
> > > >
> > > > **Q10**: Please explain whether the GenAI-based solution proposed by the authors, if it achieves similar functionality, still maintains advantages in terms of software size and simulation speed.
> > > >
> > > > **A10**: The GenAI-based solution we propose, GenNet, not only achieves similar functionality to existing simulators but also offers distinct advantages in terms of software size and simulation speed.
> > > >
> > > > First, GenNet has a compact software size of just 34,262 KB, making it more lightweight than many traditional simulators. This smaller footprint facilitates easier deployment and integration into various systems.
> > > >
> > > > Regarding simulation speed, GenNet demonstrates superior performance, as shown in Table 2 of our paper. When compared to simulators like CityFlow and MATSim for mobile user behavior simulation, and Matlab and OMNet++ for wireless network transmission simulation, GenNet exhibits significant speedup due to its parallel processing capabilities and the use of data-driven generative AI methods. These features enable GenNet to handle large-scale simulations more efficiently, achieving faster results without compromising accuracy.
> > > >
> > > > **Q11**: System-level simulation platforms use a discrete-time-driven approach for computation. During experiments, they do not require real-time waiting or the processing of actual data packets. Consequently, simulation platforms typically operate faster than real networks. This is also why many experiments are conducted in DTN environments rather than on real networks. The literature provided by the authors states that the implementation of DTN, unlike simulators, can forgo complex mechanisms, allowing it to operate faster than real networks. This contradicts the authors' claim that simulators are slower than real networks.
> > > >
> > > > **A11**: We appreciate the reviewer’s comment. However, the assertion that system-level simulation platforms typically operate faster than real networks is only valid under certain conditions, particularly when simplified models and idealized assumptions are used.
> > > >
> > > > While system-level simulators (e.g., NS-2, NS-3, OMNet++) do process virtual packets without needing real-time waiting or actual data packet handling, this simplification does not lead to faster simulations when detailed and realistic network behaviors are considered. These simulators often rely on pre-defined mechanisms such as congestion control algorithms and queuing policies, which can generate performance metrics (e.g., throughput, delay, loss rate) but only for simplified, static network models. The complexity of simulating real-world systems, especially with dynamic and evolving network conditions, results in significant computational overhead, making traditional simulators 3-4 orders of magnitude slower than real-time [1]. This slow execution is a direct result of the tight coupling and intricate design of packet-level simulators, which require processing many network protocols and layers, leading to inefficiencies.
> > > >
> > > > In contrast, GenNet is designed to handle real-time network optimizations and large-scale simulations with high accuracy and efficiency. GenNet combines parallel processing with generative AI methods to simulate dynamic mobile networks at a much faster rate than traditional simulators. As shown in Table 2 of our paper, GenNet demonstrates significant speedup and computational efficiency, benefiting from advanced AI-based simulation techniques and parallelization, without sacrificing the necessary complexity for realistic network modeling.
> > > >
> > > > Therefore, while DTN may be faster in simplified, static cases, traditional network simulators—especially those that seek to model real-world complexities—are inherently slower than actual networks. Our approach with GenNet overcomes this limitation by using advanced data-driven techniques that achieve both faster speeds and more realistic outcomes.
> > > >
> > > > [1] Linbo Hui, Mowei Wang, Liang Zhang, Lu Lu, and Yong Cui. Digital twin for networking: A data-driven performance modeling perspective. IEEE Network, 2022.

---

### Official Review · Reviewer_2yG7 · 2024-11-03

**Soundness:** 3
**Presentation:** 2
**Contribution:** 2
**Rating:** 3
**Confidence:** 5

**Summary:**

The goal of this paper is to develop a network simulation for wireless mobile systems. The simulator is based on generative AI techniques (like GANs and auto-encoders), and it focuses on problems of reinforcement learning in wireless systems. The authors discuss the simulator and benchmark its performance (including simulation time) against state-of-the-art.

**Strengths:**

+ The wireless community has a need for realistic simulators.
+ The open source nature of this simulator can help in adoption across the community.
+ The simulator seems to perform well in terms of execution time.

**Weaknesses:**

- This work does not contribute much to the field of machine learning. It uses off-the-shelf techniques to simulate a wireless system and does not really advance any of the tools used (e.g., auto-encoders, reinforcement learning, etc.). It is perhaps more appropriate for wireless conferences like SIGCOMM or GLOBECOM where the core contribution can lie in wireless and not in AI/ML.
- The authors do not provide technical justification on the choice of the various collection of machine learning techniques that they used. It seems that those choices are arbitrary and not based on technical reasons or arguments stemming from the wireless physics or the simulator.
- Reinforcement learning has not yet made it into real-world wireless systems, hence, it is questionable on whether the focus on this approach will allow the simulator to be used in real-world systems. For instance, you focus only on reinforcement learning, but that has not yet been adopted in standards like 3GPP or others. Current wireless networks use heuristics for optimization and may use ML techniques like auto-encoders for transceiver design. RL, while meaningful, cannot be the only approach used for performing evaluation of wireless networks.
- The modeling of the base stations and wireless links is very simplified and may not follow the real-world physics of wireless propagation environments.
- The utility function in (1) is arbitrary and based on very heuristic weighting. Using scalarization in multi-objective optimization is not very effective, and the authors must justify this choice. Moreover, the various terms used in this utility are not realistic wireless metrics, and they seem oversimplified.

**Questions:**

- How can you justify that the optimization problem in (1) can indeed work for real-world systems like 5G or 6G? How can you handle the realistic propagation environment characteristics like interference?
- Can you explain what are the contributions of this work to the field of AI or machine learning?
- The results in the experiments do not show a smooth behavior, can you explain why?
- How many mobile users and base stations can your simulator support?
- Can you comment on scalability?

---

> ### Author Response · Authors · 2024-11-26
>
> We thank the reviewer for recognizing our paper’s strengths and giving constructive comments. We would like to address the comments and questions below.
>
> **Q1**: How can you justify that the optimization problem in (1) can indeed work for real-world systems like 5G or 6G?
>
> **A1**: The optimization problem defined in equation (1) involves variables of base station transmission power, antenna tilt, and antenna azimuth that are directly applicable to real-world systems. The optimization objectives focus on coverage, throughput, and energy consumption, which are important performance metrics for 5G and 6G networks. These variables and objectives are representative of real-world challenges in optimizing modern mobile networks.
>
> **Q2**: How can you handle the realistic propagation environment characteristics like interference?
>
> **A2**: For interference modeling, we use Physics-informed Electromagnetic Field Network (PEFNet) which utilizes a data-driven refinement phase, which learns from measured data to account for residual errors, including interference effects. By integrating a knowledge-driven estimate with data-driven corrections, this approach captures both deterministic physical properties and stochastic elements like interference, leading to improved accuracy in modeling realistic propagation environments.
>
> **Q3**: Can you explain what are the contributions of this work to the field of AI or machine learning?
>
> **A3**: We present a platform for multi-objective optimization in mobile networks, which also includes an interface for reinforcement learning integration. This contribution significantly advances the capabilities for optimizing mobile network environments, providing a testbench for the application and evaluation of various reinforcement learning algorithms in this domain.
>
> **Q4**: The results in the experiments do not show a smooth behavior, can you explain why?
>
> **A4**: The analysis of the experimental results is provided in the appendix. The suboptimal experimental outcomes are mainly due to the complexity of the optimization scenario. Existing multi-objective algorithms struggle to effectively extract relevant information from large-scale base stations, and their update mechanisms may not be suitable for addressing the current multi-objective optimization challenges.
>
> **Q5**: How many mobile users and base stations can your simulator support?
>
> **A5**: In our simulator, the simulated mobile network includes 183 base stations and 8,000 mobile users.

---

### Official Review · Reviewer_8ibP · 2024-11-09

**Soundness:** 3
**Presentation:** 2
**Contribution:** 2
**Rating:** 5
**Confidence:** 3

**Summary:**

The authors developed GenNet—a generative AI-driven mobile network simulator to replicate mobile users, basestations, and wireless environments. GenNet features a tailor-made API explicitly designed for reinforcement learning environments, enabling researchers to finely adjust network parameters such as tilts, azimuth, and transmitting power. The simulator is then used to understand the performance gains of multi-objective optimization algorithms.

**Strengths:**

The authors proposed a generative AI based network planning and optimization tool. The tool appears to have various features as mentioned in Table-1 of the paper and a comparison is presented  with various state-of-the-art research works.

**Weaknesses:**

My main concerns are as follows:
1. Literature review is not comprehensive, especially related to multi-objective optimization in cellular networks, such as
[a] Multi-objective energy efficient resource allocation and user association for in-band full duplex small-cells
[b] Multi-objective optimization for spectrum and energy efficiency tradeoff in IRS-assisted CRNs with NOMA
[c] Generalized Multi-Objective Reinforcement Learning with Envelope Updates in URLLC-enabled Vehicular Networks
Therefore, it is not clear why multi-objective reinforcement learning techniques are beneficial compared to multi-objective optimziation techniques and how envelope outperforms other RL solutions. The authors should include a brief comparison of multi-objective optimization techniques versus multi-objective reinforcement learning approaches in the context of cellular networks. Additionally, a discussion on how Envelope compares to other RL solutions should be included.
2. The presented results are not comprehensive and it is hard to gain any insights in terms of the computation or time complexity or dimension of the Generative AI based proposed simulator. The runtime comparisons between GenNet and traditional simulators can be included, and the authors should elaborate how the simulator's performance scales with network size.
3. The considered optimization problem is rather simple as it has no non-convex quality of service rate constraints
4. the reliability of the ray tracing results and how well they match with the platforms reported in Table-1. Quantitative comparisons between their ray tracing results and those from established platforms should be included
5. Constraint violation probability should be included in the results.
6. MORL algorithms can be easily implemented using OpenAIGym, is there an interface available for OpenAI Gym. Discuss the advantages of their custom API compared to using OpenAI Gym, and explain if there are plans to provide compatibility with OpenAI Gym in future versions of GenNet.

**Questions:**

My main concerns are as follows:
1. Literature review is not comprehensive, especially related to multi-objective optimization in cellular networks, such as
[a] Multi-objective energy efficient resource allocation and user association for in-band full duplex small-cells
[b] Multi-objective optimization for spectrum and energy efficiency tradeoff in IRS-assisted CRNs with NOMA
[c] Generalized Multi-Objective Reinforcement Learning with Envelope Updates in URLLC-enabled Vehicular Networks
Therefore, it is not clear why multi-objective reinforcement learning techniques are beneficial compared to multi-objective optimziation techniques and how envelope outperforms other RL solutions. The authors should include a brief comparison of multi-objective optimization techniques versus multi-objective reinforcement learning approaches in the context of cellular networks. Additionally, a discussion on how Envelope compares to other RL solutions should be included.
2. The presented results are not comprehensive and it is hard to gain any insights in terms of the computation or time complexity or dimension of the Generative AI based proposed simulator. The runtime comparisons between GenNet and traditional simulators can be included, and the authors should elaborate how the simulator's performance scales with network size.
3. The considered optimization problem is rather simple as it has no non-convex quality of service rate constraints
4. the reliability of the ray tracing results and how well they match with the platforms reported in Table-1. Quantitative comparisons between their ray tracing results and those from established platforms should be included
5. Constraint violation probability should be included in the results.
6. MORL algorithms can be easily implemented using OpenAIGym, is there an interface available for OpenAI Gym. Discuss the advantages of their custom API compared to using OpenAI Gym, and explain if there are plans to provide compatibility with OpenAI Gym in future versions of GenNet.

---

> ### Author Response · Authors · 2024-11-26
>
> We thank the reviewer for recognizing our paper’s strengths and giving constructive comments. We would like to address the comments and questions below.
>
> **Q1**: It is not clear why multi-objective reinforcement learning techniques are beneficial compared to multi-objective optimization techniques and how envelope outperforms other RL solutions.
>
> **A1**: Multi-objective reinforcement learning (MORL) is particularly well-suited for dynamic, large-scale environments such as mobile networks, where objectives need to be balanced in real time and can change over time. In contrast, multi-objective optimization (MOO) excels in static, well-defined scenarios with a clear set of objectives, where detailed system modeling and offline computations can be effectively applied.
>
> MOO is effective for optimizing objectives such as energy efficiency and spectrum efficiency in scenarios that involve well-structured, fixed problems. It relies on mathematical frameworks like ε-constraint methods and iterative block coordinate descent to solve trade-offs between objectives. However, MOO's reliance on offline computation, system modeling, and iterative techniques makes it less adaptable to dynamic, real-time conditions, particularly in large-scale, evolving networks. This limits its application in mobile networks, where network conditions and user demands are constantly changing.
>
> On the other hand, MORL introduces a learning-based approach that can dynamically adapt to changing network conditions and user preferences without requiring retraining. By using methods such as envelope-based learning and generalized Bellman equations, MORL optimizes trade-offs like reliability, latency, and throughput in real time. envelope-based learning, in particular, allows MORL to efficiently handle multiple conflicting objectives by learning a generalized policy that balances these objectives without reducing them to a single scalarized metric. This flexibility makes MORL highly effective for dynamic networks, where conditions are unpredictable and decisions need to be made on the fly.
>
> Unlike traditional reinforcement learning (RL), which typically optimizes a single cumulative reward, MORL is designed to handle multi-objective problems by modeling and learning the trade-offs between objectives. Techniques like hindsight experience replay and convex envelope optimization further enhance MORL's ability to explore diverse preferences and learn optimal policies that perform well under various scenarios. These features make MORL an ideal solution for next-generation cellular networks, where the ability to balance conflicting goals such as ultra-reliable low-latency communication (URLLC), energy efficiency, and seamless connectivity is critical.
>
> In summary, while MOO is effective for static, well-defined problems that require detailed modeling, MORL offers a more adaptive and flexible solution for the dynamic and complex environments of modern mobile networks.
>
> **Q2**: The presented results are not comprehensive and it is hard to gain any insights in terms of the computation or time complexity or dimension of the Generative AI based proposed simulator.
>
> **A2**: Thank you for your comment. We have provided a detailed discussion of the runtime comparisons between GenNet and traditional simulators in Table 2: Simulation Efficiency of GenNet and Existing Environments. This table presents a comprehensive analysis of the simulation efficiency, including tasks related to mobile user behaviors and wireless transmission.
>
> Our results show that for both tasks, GenNet's computational efficiency significantly outperforms traditional simulators. Specifically, the simulation time is reduced by more than an order of magnitude—over 10 times faster—compared to conventional simulators. This reduction in runtime proves GenNet's better performance in handling complex simulations, particularly in large-scale, dynamic network scenarios, where traditional simulators often struggle with scalability and efficiency.

---

> > ### Comment · Reviewer_8ibP · 2024-11-29
> >
> > Thanks for clarifying the comments. Most of them are addressed. I would suggest to clarify the state-of-the-art by adding some relevant works in the area and highlighting how the proposed method can be useful for existing research works, such as.
> >
> > [a] A generalized algorithm for multi-objective reinforcement learning and policy adaptation." Advances in neural information processing systems 32 (2019).
> >
> > [b]Towards energy-efficient autonomous driving: A multi-objective reinforcement learning approach." IEEE/CAA Journal of Automatica Sinica 10.5 (2023): 1329-1331.
> >
> > [c]Generalized Multi-Objective Reinforcement Learning with Envelope Updates in URLLC-enabled Vehicular Networks." arXiv preprint arXiv:2405.11331 (2024).
> >
> > [d] An improved multi-objective deep reinforcement learning algorithm based on envelope update." Electronics 11.16 (2022): 2479.
> >
> > It would be helpful to know that reducing runtime doesnot effect significantly on the performance quality and convergence of the algorithm.
> >
> > Furthermore, it is not accurate to say that incorporating constraints like KKT conditions/Lagrangian would ensure there is no violation. The authors should read relevant research in this and make a more concrete justification about handling the quality of service constraints.
> > [a] DC3: A learning method for optimization with hard constraints
> > [b] Power control with QoS guarantees: A differentiable projection-based unsupervised learning framework

---

> > > ### Author Response · Authors · 2024-12-04
> > >
> > > **Q7**: I would suggest to clarify the state-of-the-art by adding some relevant works in the area and highlighting how the proposed method can be useful for existing research works.
> > >
> > > **A7**: As the relevant works you mentioned, reference [1] introduces a multi-objective reinforcement learning (MORL) algorithm, with subsequent studies [2], [3], and [4] building upon this foundational approach. However, all of these studies are constrained to a single-agent framework, and the optimization scenarios they address remain relatively simple, mainly focusing on applications within vehicular networks. While these works contribute valuable insights, they do not explore the challenges associated with scaling multi-objective reinforcement learning to more complex and large-scale environments.
> > >
> > > In contrast, our work extends the approach from [1] by transitioning from a single-agent to a multi-agent framework. We also address significantly more complex optimization scenarios, such as those involving hundreds of base stations. This shift to a multi-agent setting introduces new challenges, including the coordination among agents and the management of larger-scale optimization problems, which are not covered in the existing literature.
> > >
> > > [1] A generalized algorithm for multi-objective reinforcement learning and policy adaptation." Advances in neural information processing systems 32 (2019).
> > >
> > > [2]Towards energy-efficient autonomous driving: A multi-objective reinforcement learning approach." IEEE/CAA Journal of Automatica Sinica 10.5 (2023): 1329-1331
> > >
> > > [3]Generalized Multi-Objective Reinforcement Learning with Envelope Updates in URLLC-enabled Vehicular Networks." arXiv preprint arXiv:2405.11331 (2024).
> > >
> > > [4] An improved multi-objective deep reinforcement learning algorithm based on envelope update." Electronics 11.16 (2022): 2479.
> > >
> > > **Q8**: It is not accurate to say that incorporating constraints like KKT conditions/Lagrangian would ensure there is no violation.
> > >
> > > **A8**: Thank you for your comments and suggestions. It is true that in our work, we primarily focus on optimization methods based on varying input preferences in multi-objective scenarios, rather than specifically addressing constraints. Therefore, our research does not propose methods for effectively handling complex constraint problems, such as QoS constraints. We acknowledge that handling such constraints is an important challenge, but it is not the focus of our current work.
> > >
> > > As you mentioned, incorporating KKT conditions or Lagrangian multipliers could help manage constraints in some cases. However, our work does not delve into solving these constraints. Instead, our primary focus is on optimizing strategies, particularly how to make optimization decisions under different input preferences, without deeply addressing the constraint handling.

---

> ### Author Response · Authors · 2024-11-26
>
> **Q3**: The considered optimization problem is rather simple as it has no non-convex quality of service rate constraints.
>
> **A3**: Thank you for your comment. While the optimization problem may appear simple, we would like to clarify that when non-convex quality of service rate constraints are introduced, the problem can be reformulated as a multi-objective optimization problem using methods like the Karush-Kuhn-Tucker (KKT) conditions.
>
> In this case, the inequality constraints are incorporated into the optimization framework, and the problem can be solved by optimizing the trade-offs between multiple objectives, while respecting the imposed constraints. The KKT conditions provide the necessary conditions for optimality under inequality constraints, and the Pareto front can then be used to evaluate and select the best solution based on the desired balance between conflicting objectives.
>
> This approach allows the model to not only learn optimal strategies based on different input preferences but also to respect the constraints and provide solutions that effectively balance the trade-offs between multiple objectives.
>
> **Q4**: The reliability of the ray tracing results and how well they match with the platforms reported in Table-1.
>
> **A4**: We use the PEFNet to simulate ray tracing. The reliability of PEFNet, is supported by the performance metrics from experiments in [1]. In the RadioMapSeer dataset, PEFNet achieved an R2 value of 0.9731, and in the RSRPSet dataset, it reached an R2 value of 0.9314. These high R2 values, which are very close to 1, indicate a very good fit between our model's predictions and the actual path loss data. This demonstrates the accuracy and reliability of PEFNet in estimating path loss.
>
> [1] Fenyu Jiang, Tong Li, Xingzai Lv, Hua Rui, and Depeng Jin. Physics-informed neural networks for path loss estimation by solving electromagnetic integral equations. IEEE Transactions on Wireless Communications, 2024.
>
> **Q5**: Constraint violation probability should be included in the results.
>
> **A5**: We would like to clarify that our simulator has action constraints, with the multi-objective optimization algorithm reading the action bounds directly from the simulator. Thus, the action constraints are always satisfied, and no violations occur. Additionally, for constraints on the optimization objectives, we show the multi-objective optimization results on a pareto front and select solutions that meet the constraints directly from the front. Therefore, constraint violations are not observed in our results.
>
> **Q6**: MORL algorithms can be easily implemented using OpenAIGym, is there an interface available for OpenAI Gym?
>
> **A6**: Thank you for your comment regarding the use of OpenAI Gym for implementing MORL algorithms. We currently do not have an interface available for OpenAI Gym. We would like to clarify that our approach is compatible with the majority of the interfaces supported by Gym. Additionally, our framework allows users to define custom interfaces, giving them the flexibility to specify the outputs they wish to obtain. In the future, we aim to achieve full compatibility with Gym, providing a seamless integration for MORL algorithm implementations.

---

### Meta-Review · Area_Chair_LkkQ · 2024-12-08

**Metareview:**

This paper develops a network simulator for multi-objective optimization using generative AI and establishes a reinforcement learning interface. Based on a grey-box approach, the paper constructs modules for user mobility, wireless environment, and models base stations and network performance. The authors employ various generative algorithms to simulate user movement, service traffic, and wireless propagation models. Based on this simulator, the authors conducted reinforcement learning experiments to optimize base station coverage, throughput, and energy efficiency. This work does not contribute much to the field of machine learning. It uses off-the-shelf techniques to simulate a wireless system and does not really advance any of the tools used (e.g., auto-encoders, reinforcement learning, etc.).

**Additional Comments On Reviewer Discussion:**

NA

---

### Decision · Program_Chairs · 2025-01-22

Reject